# Generalization Bounds for GNNs in Transductive Node Classification: A View from Optimal Transport

## Abstract

The generalization ability of graph neural networks (GNNs) remains insufficiently understood, especially for node classification where node embeddings are inherently dependent on the entire graph structure. In this work, we establish new generalization error bounds for GNNs in the transductive node classification setting. Building on distribution-free transductive learning theory, we derive global and class-wise bounds expressed in terms of the Wasserstein distance of node features' distribution. Our analysis reveals how the GNN aggregation process transforms representation distributions and enables rigorous control of the generalization gap. We further specialize our results to the case of Simple Graph Convolution, providing explicit spectral characterizations of the bound. Empirical evaluations across homophilic and heterophilic benchmark datasets confirm that the proposed bounds accurately capture generalization behavior. These results advance the theoretical understanding of GNNs by providing the first Wasserstein-based generalization guarantees tailored to node classification.

## 1 Introduction

Graph neural networks (GNNs) achieve strong empirical performance across domains on graph-structured data, yet their generalization behavior remains insufficiently characterized (Tang & Liu, 2023). These challenges are particularly evident in node classification, where GNNs induce dependencies among node representations based on the graph structure. This interdependence breaks the standard inductive assumption that given data points are independent and identically distributed (i.i.d.) from underlying data distribution. Since the i.i.d. assumption underlies much of classical learning theory and many generalization bounds (Bartlett & Mendelson, 2002; Von Luxburg & Schölkopf, 2011; Chuang et al., 2021), their direct application to node classification becomes challenging.

Some previous works provide theoretical generalization error bounds for GNNs on the node classification task (Scarselli et al., 2018; Verma & Zhang, 2019; Zhang et al., 2020; Zhou & Wang, 2021; Garg et al., 2020). However, these analyses typically rely on a surrogate independence assumption. A common strategy is to decompose a single graph into a collection of subgraphs and treat them as independent samples, thereby enabling the use of classical generalization tools. While mathematically convenient, this abstraction does not faithfully capture the dependencies presented in real-world graphs or in practical GNNs training.

Transductive learning offers a principled alternative error bound without relying on surrogate independence. In this setting, the entire dataset is fixed in advance, and while only training data is labeled, the features of test data are also accessible. Several studies derive the upper bound of transductive Rademacher complexity to develop transductive generalization bounds for GNNs (Oono & Suzuki, 2020b; Tang & Liu, 2023; Esser et al., 2021). Despite being tailored to the transductive framework, these results have limited applicability and practical utility. Many bounds hold only under restrictive assumptions, such as a fixed model structure (Oono & Suzuki, 2020b; Cong et al., 2021; Esser et al., 2021) or a designated training algorithm (Tang & Liu, 2023; Cong et al., 2021), which prevents extension to broader GNN architectures or optimization procedures. Furthermore,

most approaches rely on Rademacher complexity (Bartlett & Mendelson, 2002), which is difficult to compute and often misaligned with empirical observations (Jiang et al., 2019; Chuang et al., 2021).

In parallel, Chuang et al. (2021) introduces a representation-based margin bound that incorporates $k$-variance (Solomon et al., 2022) via optimal transport, aiming to better align theoretical guarantees with empirical performance by quantifying structure in the learned feature distribution. However, their analysis relies on an i.i.d. sampling assumption and thus does not account for the transductive setting. Consequently, the bound is not directly applicable to GNN-based node classification on a fixed graph, while it can be applied to graph classification tasks (Li et al., 2025).

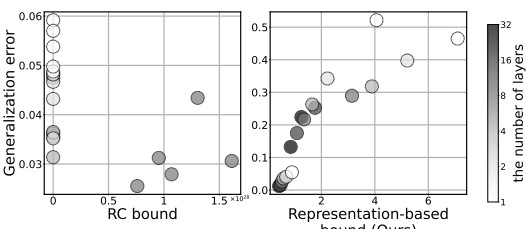

Motivated by these challenges, we propose two new generalization error bounds for GNNs in the transductive node classification setting. Both bounds are formulated in terms of the 1-Wasserstein distance between feature distribu-

Figure 1: Correlation between the generalization error of GNNs and two types of generalization bounds. The Rademacher complexity (RC) bound shows weak correlation with the generalization error and instead aligns more closely with depth, whereas our proposed representation-based bound exhibits strong correlation with the actual generalization error regardless of the depth of GNN.

tions, with one capturing the discrepancy between training and test sets, and the other quantifying intra-class concentration and inter-class separation of learned feature distribution. These bounds are directly computable from observed features and model parameters, making them broadly applicable across GNN architectures without requiring specific algorithms. We further provide a theoretical analysis that reveals the relation between the Wasserstein distance of feature distributions and the depth of GNNs. This analysis shows that our bound offers a unified perspective on existing oversmoothing studies from a generalization perspective.

We validate our bounds through extensive experiments on homophilic and heterophilic benchmarks. The results show that our bounds exhibit strong rank correlation with the empirical generalization gap across diverse GNN architectures and datasets, substantially outperforming the Rademacher complexity (RC) bound, which is often misaligned with empirical trends, as shown in Figure 1. These findings confirm that our Wasserstein-based bounds not only provide new theoretical insights but also reliably capture generalization behavior in practice.

## 2 RELATED WORK

**Representation-based generalization bounds** Representation-based complexity measures have been proposed as alternatives to classical notions such as VC-dimension (Vapnik & Chervonenkis, 2015) or norm-based complexity (Bartlett et al., 2017). Natekar & Sharma (2020) introduced a representation-based measure that demonstrated stronger predictive power of generalization compared to traditional measures in the Predicting Generalization in Deep Learning competition (Jiang et al., 2020). Chuang et al. (2021) developed a margin-based bound incorporating the $k$-variance (Solomon et al., 2022), derived from optimal transport, to account for structural properties of learned feature distributions. Li et al. (2025) extended this line of work to graph classification tasks, characterizing the representation space of graphs through the expressivity of GNN models.

**Generalization bounds for node classification** Generalization analysis for node-level prediction with GNNs faces unique challenges due to dependencies among node representations. Some early works established bounds by decomposing a single graph into subgraphs and treating them as independent samples (Scarselli et al., 2018; Verma & Zhang, 2019; Zhang et al., 2020; Zhou & Wang, 2021; Garg et al., 2020). Other works studied distribution-free transductive learning algorithms (El-Yaniv & Pechyony, 2009). For example, Oono & Suzuki (2020b) derived bounds based on Rademacher complexity under a specific optimization procedure, Esser et al. (2021) provided bounds that explicitly depend on GNN components, and Tang & Liu (2023) and Cong et al. (2021) incorporated graph properties such as node degrees, spectral norms of weight matrices, number of layers, and training iterations into their analyses.

**Generalization bounds for graph classification** Several studies addressed generalization at the graph level. Liao et al. (2021) established PAC-Bayesian bounds relying on node degrees and spectral norms, which were refined by Ju et al. (2023) using the largest singular value of diffusion matrices. Du et al. (2019) employed neural tangent kernels to analyze infinitely wide GNNs trained with gradient descent. Karczewski et al. (2024) studied bounds for $E(n)$-equivariant GNNs. Maskey et al. (2022) showed that the generalization gap improves with the number of training samples and average graph size. Morris et al. (2023) analyzed the VC dimension of GNNs using the Weisfeiler–Leman test, and Levie (2023) derived generalization bounds in the graphon-signal space using cut distance.

## 3 PRELIMINARIES

**Generalization bound in transductive learning** Transductive learning refers to the paradigm where labeled training data and unlabeled test data are given, and the learner is required to predict the labels of the test data. Unlike standard supervised learning, where neither the features nor the labels of the test data are accessible during training, transductive learning allows access to the features of the test data at training time, while only the training data come with labels. A formal definition of a distribution-free transductive model is provided by Vapnik (2006). Consider a fixed dataset $\mathcal{D} = \{(\mathbf{x}_i, y_i)\}_{i=1}^{m+u}$ of $m+u$ data points $\mathbf{x}_i \in \mathbb{R}^F$ and labels $y_i \in \{1, \cdots, K\}$, where $K$ denotes the number of classes. Using a random permutation $\pi : \{1, \cdots, m+u\} \to \{1, \cdots, m+u\}$, the training set is determined as $\mathcal{D}_{\text{train}} = \{(\mathbf{x}_{\pi(i)}, y_{\pi(i)})\}_{i=1}^{m}$ and test set as $\mathcal{D}_{\text{test}} = \{(\mathbf{x}_{\pi(i)}, y_{\pi(i)})\}_{i=m+1}^{m+u}$. During training, the learner has access to the full unlabeled samples $\{\mathbf{x}_{\pi(i)}\}_{i=1}^{m+u}$ and the labels $\{y_{\pi(i)}\}_{i=1}^{m}$. Building generalization bound in transductive learning focuses on bounding test error $\frac{1}{u}\sum_{i=m+1}^{m+u} \ell(\mathbf{x}_{\pi(i)}, y_{\pi(i)})$ for any permutation $\pi$, where $\ell$ is the loss function. For convenience, it is common to omit the explicit notation of the permutation $\pi$ by re-indexing the data points as $\mathbf{x}_i \leftarrow \mathbf{x}_{\pi(i)}$ and $y_i \leftarrow y_{\pi(i)}$.

**Graph neural networks** Consider an undirected graph $\mathcal{G} = (\mathcal{V}, \mathcal{E})$, where $\mathcal{V}$ denotes a set of $N \in \mathbb{N}$ nodes and $\mathcal{E} \subseteq \mathcal{V} \times \mathcal{V}$ represents the edge set. Each node $i \in \mathcal{V}$ is associated with a $F$- representation vector $\mathbf{x}_i \in \mathbb{R}^F$. These representations are collectively represented by the matrix $\mathbf{X} \in \mathbb{R}^{N \times F}$. The graph structure can be encoded by a binary, symmetric adjacency matrix $\mathbf{A} \in \{0, 1\}^{N \times N}$, where $\mathbf{A}_{ij} = 1$ if an edge exists between node $i$ and $j$, and $\mathbf{A}_{ij} = 0$, otherwise.

Graph Neural Networks (GNNs) are characterized by an aggregation process that leverages edge information to capture interactions between neighboring nodes. For example, the aggregation process in Graph Convolutional Networks (GCNs) (Kipf, 2016) can be formalized as $\hat{\mathbf{A}}\mathbf{X}$, where $\hat{\mathbf{A}} := \tilde{\mathbf{D}}^{-1/2}\tilde{\mathbf{A}}\tilde{\mathbf{D}}^{-1/2}$, $\tilde{\mathbf{A}} := \mathbf{A} + \mathbf{I}$, and $\tilde{\mathbf{D}} := \text{diag}(\tilde{\mathbf{A}}\,\mathbf{1})$. The aggregation step updates each node feature as a weighted average of its neighbors' features, inducing dependencies among node features. The output of the $\ell$-th layer $\mathbf{X}^{(\ell)}$ in GCNs is calculated with additional linear transformation and non-linear activation steps, $\mathbf{X}\mathbf{W}$ and $\sigma(\cdot)$, given by:

$$\mathbf{X}^{(\ell+1)} = \sigma(\hat{\mathbf{A}}\mathbf{X}^{(\ell)}\mathbf{W}^{(\ell)}) \,,$$

where $\mathbf{X}^{(0)} = \mathbf{X}$ and $0 \leq \ell < L$. In the case of Simple Graph Convolution (SGC) (Wu et al., 2019), the non-linear activation in GCNs is removed, resulting in:

$$\mathbf{X}^{(L)} = \hat{\mathbf{A}}^L \mathbf{X}^{(0)} \mathbf{W} \,.$$

**Wasserstein distance** Given two probability distributions $\mu$ and $\nu$ on $\mathbb{R}^F$, the $p$-Wasserstein distance between $\mu$ and $\nu$ with Euclidean cost is defined as:

$$\mathcal{W}_p(\mu, \nu) := \inf_{T \in \mathcal{U}(\mu,\nu)} \left(\mathbb{E}_{(x,y)\sim T}\|x - y\|^p\right)^{1/p} \,,$$

where $\mathcal{U}(\mu, \nu)$ denotes the set of all couplings of $\mu$ and $\nu$, i.e., joint distributions $T$ on $\mathbb{R}^F \times \mathbb{R}^F$ with $\mu$ and $\nu$ as marginals. Intuitively, the Wasserstein distance measures the minimal cost of transporting mass from distribution $\mu$ to $\nu$. Throughout this work, we adopt the 1-Wasserstein distance and denote $\|\cdot\|$ as Euclidean norm.

# 4 WASSERSTEIN BOUNDS IN TRANSDUCTIVE LEARNING

In this section, given an encoder $\phi$, a dataset $\mathcal{D} = \{(\mathbf{x}_i, y_i)\}_{i=1}^{m+u}$ with an arbitrary train-test split $\pi$, we first derive the generalization error bound that depends on a given split $\pi$ in the transductive setting. We then establish the high-probability bound on the generalization gap within the transductive framework.

**Setup** We adapt the setup of Chuang et al. (2021) for the task of transductive node classification, a widely adopted scenario in graph machine learning (Yang et al., 2016; Kipf, 2016). In the transductive learning scenario, the whole graph structure $\mathbf{A}$ and all node representation $\mathbf{X}$ are available, but labels are provided only for training nodes. More formally, let $\mathcal{X}$ denote the initial node representation space, $\mathcal{Z}$ the embedding space, and $\mathcal{Y} = \{1, \cdots, K\}$ the output space. We consider a compositional hypothesis class $\mathcal{F} \circ \Phi$, with feature encoder $\Phi = \{\phi : \mathcal{X} \times (\mathcal{X}^N, \mathcal{A}) \to \mathcal{Z}\}$ and score-based classifier $\mathcal{F} = \{f = [f_1, \cdots, f_K] : \mathcal{Z} \to \mathbb{R}^K\}$. The label of node $i$ is predicted by $\arg\max_y f_y(\phi(\mathbf{x}_i; \mathbf{X}, \mathbf{A}))$. We set $\phi$ as a graph neural network (GNN), which includes an aggregation process that combines feature information from neighboring nodes based on $\mathbf{X}$ and $\mathbf{A}$, thereby making each output embedding inherently dependent on the entire graph. The margin of classifier $f$ for a node data point $(\mathbf{x}, y)$ is defined by:

$$\rho_f(\phi(\mathbf{x}; \mathbf{X}, \mathbf{A}), y) := f_y(\phi(\mathbf{x}; \mathbf{X}, \mathbf{A})) - \max_{y' \neq y} f_{y'}(\phi(\mathbf{x}; \mathbf{X}, \mathbf{A})) .$$

The node is misclassified if $\rho_f(\phi(\mathbf{x}; \mathbf{X}, \mathbf{A}), y) \leq 0$. An undirected graph $\mathcal{G}$ with node representation $\mathbf{X} \in \mathbb{R}^{N \times F}$ and random permutation $\pi$ is given. We define index sets for train and test as $\mathcal{I}_{\text{train}}^{(\pi)} := \{\pi(i)\}_{i=1}^m$ and $\mathcal{I}_{\text{test}}^{(\pi)} := \{\pi(i)\}_{i=m+1}^{m+u}$. The training and test dataset is split by the index set. We are interested in bounding the generalization gap between the zero-one loss of test set $R_u(f \circ \phi; \pi)$ and the $\gamma$-margin loss of train set $R_{m,\gamma}(f \circ \phi; \pi)$ of a model $f \circ \phi$ and permutation $\pi$, where: $R_u(f \circ \phi; \pi) := \frac{1}{u} \sum_{i \in \mathcal{I}_{\text{test}}^{(\pi)}} \mathbb{1}_{\rho_f(\phi(\mathbf{x}_i; \mathbf{X}, \mathbf{A}), y_i) \leq 0}$, and $R_{m,\gamma}(f \circ \phi; \pi) := \frac{1}{m} \sum_{i \in \mathcal{I}_{\text{train}}^{(\pi)}} \mathbb{1}_{\rho_f(\phi(\mathbf{x}_i; \mathbf{X}, \mathbf{A}), y_i) \leq \gamma}$, with $\gamma > 0$.

## 4.1 THEORETICAL ANALYSIS

We derive two transductive generalization bounds based on the Wasserstein distance: one involving the distance between the training and test encoded feature distributions, and the other involving the expected sum of Wasserstein distances between feature distributions within the same class. The first theorem allows direct computation of the error bound, while the second theorem shows that the concentration and separation of learned features previously identified as key factors for generalization bounds in the inductive setting (Chuang et al., 2021) are also crucial in the transductive setting.

To formalize the first theorem, we define the empirical distribution of node representation $\mu_{\mathcal{I}}$ for given index set $\mathcal{I}$ as $\mu_{\mathcal{I}} := \frac{1}{|\mathcal{I}|} \sum_{i \in \mathcal{I}} \delta_{\mathbf{x}_i}$, where $\delta(x) := \delta_x$ denotes the Dirac delta function. The distribution $\phi_{\#}\mu$ is the result of applying the pushforward measure operation on $\mu$ with respect to $\phi(\cdot; \mathbf{X}, \mathbf{A})$, i.e., the distribution of $\phi(x)$ when $x$ is drawn from $\mu$. Our first main result is as follows:

---

**Theorem 4.1** (Global Wasserstein bound in the transductive setting)**.** *Let $\gamma > 0$. For any random split $\pi$, and all $f \circ \phi \in F \circ \Phi$,*

$$R_u(f \circ \phi; \pi) \leq R_{m,\gamma}(f \circ \phi; \pi) + \frac{M(f, \phi)}{\gamma} \mathcal{W}_1\big(\phi_{\#}\mu_{\mathcal{I}_{\text{train}}^{(\pi)}}, \phi_{\#}\mu_{\mathcal{I}_{\text{test}}^{(\pi)}}\big) , \qquad (1)$$

*where*

$$M(f, \phi) := \max_{i,j,y} \frac{|\rho_f(\phi(\mathbf{x}_i), y_i) - \rho_f(\phi(\mathbf{x}_j), y)|}{\|\phi(\mathbf{x}_i) - \phi(\mathbf{x}_j)\|} \quad \text{for } i \in \mathcal{I}_{\text{train}}^{(\pi)}, \ j \in \mathcal{I}_{\text{test}}^{(\pi)}, \ y \in \mathcal{Y}.$$

---

Theorem 4.1 demonstrates that the generalization error is small under three conditions: 1) the distance between the feature distributions of the training and test sets is small, 2) the change rate of the margin of classifier $f$ is small, or 3) the margin of the classifier is large. Since we can access the

encoded feature of both the test as well as training in the transductive setting, we can readily obtain the generalization error bound Equation (1) by computing the Wasserstein distance between the two distribution and all possible values of $|\rho_f(\phi(\mathbf{x}_i), y_i) - \rho_f(\phi(\mathbf{x}_j), y)| / \|\phi(\mathbf{x}_i) - \phi(\mathbf{x}_j)\|$ where $i \in \mathcal{I}_{\text{train}}^{(\pi)}, j \in \mathcal{I}_{\text{test}}^{(\pi)}$ and $y \in \mathcal{Y}$.

We now introduce our second bound, which connects generalization to the class-wise feature distributions. To formalize our theorem, we define $\mathcal{I}_{\text{train},c}^{(\pi)} \coloneqq \{i \in \mathcal{I}_{\text{train}}^{(\pi)} | y_i = c\}$ and $\mathcal{I}_{\text{test},c}^{(\pi)} \coloneqq \{i \in \mathcal{I}_{\text{test}}^{(\pi)} | y_i = c\}$ for each class $c$. let $m_c^{(\pi)} \coloneqq |\mathcal{I}_{\text{train},c}^{(\pi)}|$ and $u_c^{(\pi)} \coloneqq |\mathcal{I}_{\text{test},c}^{(\pi)}|$ denoting the number of training and test samples with label $c$ respectively. We represent the second main theorem on generalization error with high probability as follows:

---

**Theorem 4.2** (Class-wise Wasserstein bound in the transductive setting). *Let $\gamma > 0$. Then, with probability at least $1 - \delta$ over the random split $\pi$, for all $f \circ \phi \in F \circ \Phi$,*

$$R_u(f \circ \phi; \pi) \le R_{m,\gamma}(f \circ \phi; \pi) + \mathbb{E}_{\pi'}\left[\sum_{c=1}^{K}\left|\frac{u_c^{(\pi')}}{u} - \frac{m_c^{(\pi')}}{m}\right|\right]$$

$$+ \sum_{c=1}^{K}\frac{M_c(f, \phi)}{\gamma}\mathbb{E}_{\pi'}\left[\frac{m_c^{(\pi')}}{m}\, \mathcal{W}_1\left(\phi_{\#}\mu_{\mathcal{I}_{\text{train},c}^{(\pi')}}, \phi_{\#}\mu_{\mathcal{I}_{\text{test},c}^{(\pi')}}\right)\right] + \varepsilon_\delta, \quad (2)$$

*where*

$$M_c(f, \phi) \coloneqq \max_{i,j}\frac{|\rho_f(\phi(\mathbf{x}_i), c) - \rho_f(\phi(\mathbf{x}_j), c)|}{\|\phi(\mathbf{x}_i) - \phi(\mathbf{x}_j)\|} \quad \text{for } i \ne j \text{ and } i, j \in \mathcal{I}_{\text{train},c}^{(\pi)} \cup \mathcal{I}_{\text{test}}^{(\pi)},$$

$$\varepsilon_\delta = \sqrt{\frac{m\,u\,\beta^2}{2\,(m + u - \frac{1}{2})}\left(1 - \frac{1}{2\max\{m, u\}}\right)^{-1}\ln\frac{1}{\delta}}, \quad \text{and} \quad \beta = \frac{1}{m} + \frac{1}{u}.$$

---

We provide a proof of Theorem 4.2 in Appendix A.2. Theorem 4.2 identifies four explicit conditions that each reduce the transductive generalization gap: 1) small expected 1-Wasserstein distance between the training and test feature distributions within each class $c$, 2) the rate of change of the margin of classifier $f$ within each class $c$ is small; 3) margin of the classifier $\gamma$ is large; or 4) the expected sum of difference in class proportions between the training and test sets across all classes is small.

A distinguishing aspect of Theorem 4.2 compared to Theorem 4.1 lies in the third term in Equation (2). The expectation term over random splits $\pi'$ in the third term highlights structural properties of the encoded features that achieve better generalization. Because each random split reassigns training and test indices, the expected sum of intra-class Wasserstein distances over random splits is close to measuring the Wasserstein distance between arbitrary subsets of features within the same class across the entire dataset. Therefore, a smaller expected value indicates that the features of class $c$ become more concentrated under the encoder $\phi$, thereby reducing the class-wise contribution to the generalization gap.

Theorem 4.2 extends the inductive bound of Chuang et al. (2021) to the transductive setting. Chuang et al. (2021) reveals that the concentration and separation of feature distributions are key factors for generalization in the inductive setting. In particular, they explain the concept of separation by showing that a larger margin in the learned model is equivalent to a larger inter-class Wasserstein distance, $\mathcal{W}_1(\phi_{\#}\mu_{\mathcal{I}_c}, \phi_{\#}\mu_{\mathcal{I}_{c'}})$. Since we adopt the same margin-based loss in our theory, this interpretation can be transferred to our theorem. Therefore, our result demonstrates that the concentration and separation of feature distributions are likewise crucial for generalization in the transductive setting.

A key distinction between the two bounds is that, while Chuang et al. (2021) use $\text{Lip}(\rho_f(\cdot, c))$, which indicates the Lipschitz constant of the class-$c$ margin, our theorem leverage $M_c(f, \phi)$ expliting access to test data features. Since $M_c(f, \phi) \le \text{Lip}(\rho_f(\cdot, c))$ the resulting bound is tighter. A further advantage of $M_c(f, \phi)$ is its exact computability for ReLU network classifiers, whereas $\text{Lip}(\rho_f(\cdot, c))$ is NP-hard to obtain and must be approximated.

Table 1: Correlation between empirical error gap and generalization bounds across datasets and GNNs. *Global* reports our bound from Theorem 4.1, while *oracle* and *approx* correspond to the class-wise bound in Theorem 4.2 with and without test labels, respectively. Table shading reflects correlation: darker for lower, lighter for higher, indicating stronger alignment with performance. The *RC bound* (Esser et al., 2021) is shown for the applicable models.

| | | Cora | CiteSeer | PubMed | Computers | Photo | Squirrel | Chameleon | Roman-empire | Amazon-ratings |
|---|---|---|---|---|---|---|---|---|---|---|
| SGC | global | 0.92 | 0.96 | 0.96 | 0.19 | 0.84 | 0.95 | 0.76 | 0.80 | 0.98 |
| | oracle | 0.89 | 0.82 | 0.98 | 0.26 | 0.81 | 0.87 | 0.62 | 0.78 | 0.93 |
| | approx | 0.87 | 0.83 | 0.98 | 0.20 | 0.81 | 0.85 | 0.58 | 0.78 | 0.91 |
| | RC bound | -0.92 | -0.97 | -0.38 | -0.06 | -0.25 | -0.94 | -0.94 | -0.56 | -0.77 |
| GCN | global | 0.82 | 0.79 | 0.82 | 0.77 | 0.79 | 0.70 | 0.48 | 0.68 | 0.93 |
| | oracle | 0.81 | 0.66 | 0.69 | 0.61 | 0.63 | 0.42 | 0.06 | 0.69 | 0.91 |
| | approx | 0.78 | 0.60 | 0.68 | 0.56 | 0.57 | 0.38 | 0.01 | 0.65 | 0.91 |
| | RC bound | 0.29 | 0.48 | 0.35 | -0.01 | -0.27 | -0.75 | -0.82 | 0.35 | 0.20 |
| GCNII | global | 0.89 | 0.82 | 0.67 | 0.88 | 0.90 | 0.18 | -0.39 | 0.68 | 0.95 |
| | oracle | 0.82 | 0.77 | 0.66 | 0.83 | 0.87 | 0.17 | -0.28 | 0.70 | 0.91 |
| | approx | 0.77 | 0.75 | 0.67 | 0.81 | 0.84 | 0.17 | -0.31 | 0.70 | 0.90 |
| GAT | global | 0.75 | 0.64 | 0.43 | 0.70 | 0.63 | 0.44 | 0.47 | 0.74 | 0.89 |
| | oracle | 0.70 | 0.62 | 0.34 | 0.70 | 0.66 | 0.63 | 0.62 | 0.74 | 0.80 |
| | approx | 0.68 | 0.58 | 0.32 | 0.70 | 0.64 | 0.62 | 0.57 | 0.68 | 0.79 |
| SAGE | global | 0.78 | 0.75 | 0.55 | 0.90 | 0.61 | 0.40 | 0.53 | 0.66 | 0.79 |
| | oracle | 0.76 | 0.88 | 0.55 | 0.87 | 0.75 | 0.31 | 0.25 | 0.68 | 0.69 |
| | approx | 0.76 | 0.88 | 0.54 | 0.86 | 0.73 | 0.33 | 0.23 | 0.68 | 0.68 |

## 5 EXPERIMENT

We conduct an empirical study to evaluate our proposed generalization bounds for the transductive node classification. To show the robustness of our theory, we use five GNN architectures, SGC (Wu et al., 2019), Graph Convolutional Network (GCN) (Kipf, 2016), GCNII (Chen et al., 2020), Graph Attention Network (GAT) (Veličković et al., 2017), and GraphSAGE (Hamilton et al., 2017) on nine datasets. We report rank correlations between our bounds and the empirical generalization gap. High correlations mean that empirical results support our bounds.

### 5.1 DATASETS AND EXPERIMENTAL SETUP

**Datasets** We validate our theory using nine datasets, comprising five homophilic and four heterophilic graphs. The homophilic datasets include Cora, CiteSeer, PubMed, Computers, and Photo (Sen et al., 2008; Yang et al., 2016; McAuley et al., 2015). For heterophilic datasets, we use Squirrel, Chameleon, Roman-empire, and Amazon-ratings (Platonov et al., 2023). Following the methodology of Platonov et al. (2023), we applied a filtering process to both Chameleon and Squirrel to prevent train-test leakage. The key statistics for these datasets are summarized in Appendix B.

**Implementation details** We follow the standard transductive learning setting (Tang & Liu, 2023), where, for each run, a random seed is used to select $30\%$ of the nodes for the training set, with the remaining $70\%$ of the nodes serving as the test set. For classifier $f$, we use one-, two-, four-MLP layers with ReLU activation function. All models are trained for $500$ iterations using the Adam optimizer. We set the hidden dimension to $64$ and the learning rate to $0.01$.

**Experimental setup** For the baseline, we use the RC bound (Esser et al., 2021). We highlight that only the RC bound can be applied as a baseline, and even to the two GNN models (SGC and GCN) among many bounds for transductive node classification. Since RC bound is derived from the hypothesis class that corresponds to GCN or SGC and other bounds derived under restrictive assumptions, such as specific architectures (Oono & Suzuki, 2020a; Cong et al., 2021) or optimization algorithms (Tang & Liu, 2023; Cong et al., 2021). To validate Theorem 4.1, we report results under *global*. We empirically found that *global* shows higher correlation when we use 0.9 percentile of the change rate among all combination sets of $(i, j, y)$, instead of using the maximum value, $M(f, \phi)$. Hence, we adopt the 0.9 percentile throughout our experiments when reporting results. We provide the results with the maximum value, $M(f, \phi)$ in Appendix D. For Theorem 4.2, we

present two variations: an *oracle*, which computes the bound with access to test labels, and an *approx*, which estimates the bound using only training data. The *oracle* validates the theory, while the *approx* illustrates a practical approximation suitable for real training scenarios. Further details for the approximation method are provided in Appendix C.

## 5.2 RESULTS

**Correlation between empirical generalization gap and theoretical bounds**   Table 1 shows the rank correlation between our proposed theoretical error bound and the empirical generalization gap. The experimental results show that our error bound is strongly correlated with the error gaps observed in experiments across various model architectures and datasets, indicating its robustness. In particular, *global* consistently attains a correlation above 0.6 across all GNN architectures on 5 out of 9 datasets: Cora, CiteSeer, Photo, Roman-empire, and Amazon-ratings. Moreover, about 78% of the reported correlations are above 0.6. These results demonstrate that our proposed error bound Theorem 4.1 is strongly supported by empirical evidence.

For Theorem 4.2, both *oracle* and *approx* show correlations above 0.7 with the empirical error gap on 7 out of 9 datasets for the SGC model, and on 6 out of 9 datasets for the GCNII model. These results provide empirical justification for our proposed Theorem 4.2.

**Comparison with RC bound**   Our results consistently outperform the RC bound across all datasets and GNN architectures, showing a higher correlation with empirical error bounds. Notably, the RC bound exhibits a strong negative correlation on certain datasets. This behavior arises because the RC bound grows exponentially with model depth and approaches infinity beyond a certain depth, resulting in a mismatch with empirical results, as shown in Table 1 and Figure 1.

## 6 ANALYZING DEPTH-INDUCED TRADE-OFFS IN GNN GENERALIZATION

In Section 4, we introduce a generalization error bound for node classification based on the Wasserstein distance. In particular, Theorem 4.2 shows that within-class concentration and between-class separation is key factor on the generalization error bound. Aggregation in GNNs makes neighboring features more similar. When applied repeatedly, this process eventually leads to concentrated features for all nodes, a phenomenon known as oversmoothing (Li et al., 2018; Keriven, 2022; Chamberlain et al., 2021). Thus, greater depth tightens the bound from the perspective of concentration but loosens it from the perspective of separation.

Motivated by this trade-off induced by aggregation, we study how GNN depth influences generalization error using our theorems. First, we theoretically analyze how repeated aggregation (depth) affects the Wasserstein distance using Simple Graph Convolution (SGC) (Wu et al., 2019). We then validate our theory by empirically observing how the Wasserstein distance evolves with depth, and also explore the trends of the bound and the generalization error to explore the aforementioned trade-off.

### 6.1 THEORETICAL ANALYSIS

In this section, we focus on the 1-Wasserstein distance between SGC-encoded feature distributions of two arbitrary node subsets to study how aggregation reshapes features as depth increases. Intuitively, since aggregation makes features closer, Wasserstein distance between feature distributions is expected to shrink with increasing aggregation depth. We establish the theorem that rigorously confirms our intuition.

To formalize the theorem, we set the SGC encoder with the propagation depth $\ell \in \mathbb{N}$ as $\phi^{(\ell)}(\mathbf{x}_i; \mathbf{X}, \mathbf{A}) \coloneqq (\hat{\mathbf{A}}^\ell \mathbf{X})_i$, where $(\hat{\mathbf{A}}^\ell \mathbf{X})_i$ is the SGC encoded feature of node $i$ at depth $\ell$, and $d(\mathbf{x}_i) = \sqrt{\tilde{d}_i}$, where $\tilde{d}_i \coloneqq (\tilde{\mathbf{D}})_{ii}$.

**Theorem 6.1** (Wasserstein distance of encoded features of SGC). *Let $\mathcal{S}, \mathcal{T} \subseteq [N]$ be any nonempty index subsets. Then, for every propagation depth $\ell \in \mathbb{N}$,*

$$\mathcal{W}_1\big(\phi_\#^{(\ell)}\mu_\mathcal{S}, \phi_\#^{(\ell)}\mu_\mathcal{T}\big) \;\le\; C_1\big(\mathbf{X}, \hat{\mathbf{A}}\big)\,\mathcal{W}_1\big(d_\#\mu_\mathcal{S},\, d_\#\mu_\mathcal{T}\big) \;+\; C_2(\mathbf{X})\,\rho_\perp(\hat{\mathbf{A}})^\ell, \qquad (3)$$

*where $\rho_\perp(\hat{\mathbf{A}}) \coloneqq \max_{k \in \{2,\dots,N\}}\{|\lambda_k(\hat{\mathbf{A}})| \;:\; \lambda_1(\hat{\mathbf{A}}) = 1\} \in [0,1)$ is the nontrivial spectral radius, and $C_1(\mathbf{X}, \hat{\mathbf{A}})$, $C_2(\mathbf{X})$ are finite constants depending only on $(\mathbf{X}, \hat{\mathbf{A}})$.*

The proof of Theorem 6.1 is provided in Appendix A.3. For a given graph $\mathcal{G}$ and the SGC encoder, Theorem 6.1 identifies an upper bound of the Wasserstein distance. The theorem establishes that for any two node sets encoded by the SGC encoder, the Wasserstein distance between their features decomposes into two terms: one depend to the Wasserstein distance of the degree distribution while independent to the depth $\ell$, and another expressed as an exponential in depth $\ell$ with base $\rho_\perp(\hat{\mathbf{A}})$. That means, when depth increases, the Wasserstein distance decays exponentially with rate $\rho_\perp(\hat{\mathbf{A}}) \in [0,1)$, to converge to the Wasserstein distance of the degree distribution.

## 6.2 EMPIRICAL STUDIES

In this section, we empirically analyze how concentration and separation evolve as SGC depth increases and connect these trends to our error bound in Theorem 4.1, and Theorem 4.2 and the empirical generalization gap. We vary the number of layers from 1 to 64, and calculate six measures: 1) Wasserstein distance between train and test feature distribution ($\mathcal{W}$); 2) average value of expected Wasserstein distance for each class ($\mathcal{W}_C$); 3) average value of Wasserstein distance for different class ($\mathcal{W}_S$); 4) global bound; 5) oracle bound; and 6) generalization error. In particular, intra-class concentration $\mathcal{W}_C$ and inter-class separation $\mathcal{W}_S$ are computed as

$$\frac{1}{K}\sum_{c=1}^{K}\mathbb{E}_{\pi'}\left[\frac{m_c^{(\pi')}}{m}\,\mathcal{W}_1\left(\phi_\#\mu_{\mathcal{I}_{\text{train},c}^{(\pi')}},\,\phi_\#\mu_{\mathcal{I}_{\text{test},c}^{(\pi')}}\right)\right] \quad \text{and} \quad \frac{2}{K(K-1)}\sum_{c_1 \neq c_2}\mathcal{W}_1\big(\phi_\#\mu_{\mathcal{I}_{c_1}},\,\phi_\#\mu_{\mathcal{I}_{c_2}}\big)\;,$$

respectively. Motivated by the intuition that the early behavior of intra-class concentration and inter-class separation depends on the homophilic ratio of the graph, we consider a homophilic dataset, Cora, and a heterophilic dataset, Amazon-ratings. All experiments are independently repeated five times, and we report the mean and standard deviation across all runs.

The results are shown in Figure 2. In both datasets, all Wasserstein distances decay exponentially, which aligns with our theory. In particular, for the Cora dataset, $\mathcal{W}$ and $\mathcal{W}_C$ decay more rapidly and tend to converge at shallower depths, whereas $\mathcal{W}_S$ does not converge even at depth 64. As a result, at shallow depths the bounds decrease due to the rapid decay of $\mathcal{W}$ and $\mathcal{W}_C$, while at larger depths the decrease of $\mathcal{W}_S$ dominates, causing the bounds to increase. The actual generalization error exhibits a similar trend to the bounds, decreasing initially and then rising again. In the case of a heterophilic dataset, $\mathcal{W}$ and $\mathcal{W}_C$ converge more slowly, consistently decreasing to depth 64. Consequently, after a certain depth, the bounds begin to decrease, and the actual generalization error is also observed to decrease with increasing depth. Overall, the two types of datasets exhibit opposite trends at greater depths, indicating that the optimal choice of depth should consider the homophilic ratio.

## 6.3 CONNECTIONS TO PRIOR OVERSMOOTHING WORK

We found that, interpreting Wasserstein distance in Theorem 4.2 through the analysis provided in Theorem 6.1 offers a unifying view of existing oversmoothing studies and suggests directions for improvement. More specifically, we connect our results to two lines of prior research on oversmoothing.

The first line of work (Cai & Wang, 2020; Li et al., 2018; Oono & Suzuki, 2020a) establishes that node features in GCNs converge to degree-scaled feature space as aggregation is repeatedly applied. Furthermore, Cai & Wang (2020); Oono & Suzuki (2020a) define an oversmoothing measure that quantifies the extent of convergence among node features and show that it decays exponentially when the depth increases. Since the expectation of the Wasserstein distance reflects the degree of concentration among node features, it is conceptually equivalent to the oversmoothing measure. Therefore, our results in Theorem 6.1 show the aligned results with Cai & Wang (2020); Li et al.

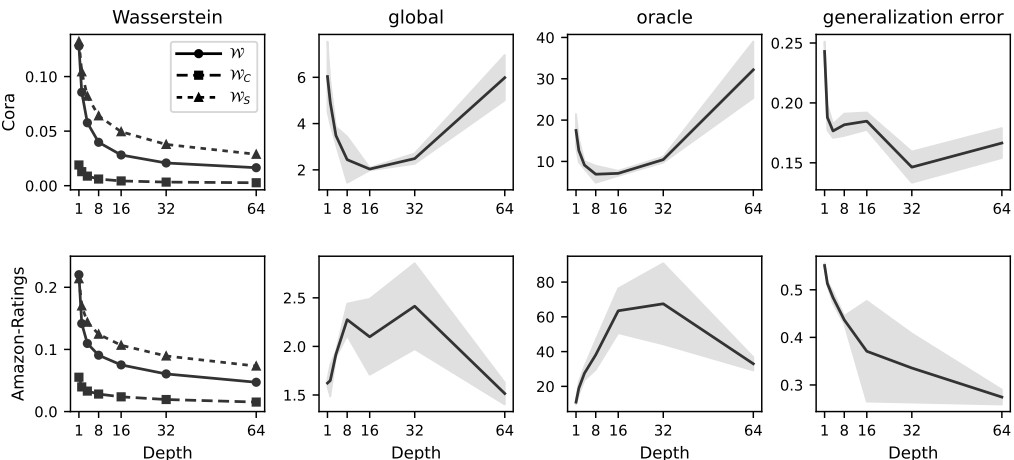

Figure 2: Variation of different measures as a function of GNN layers for Cora and Amazon-Ratings datasets. We report the trajectories of three Wasserstein distances, two generalization bounds, and empirical generalization gaps.

(2018); Oono & Suzuki (2020a), where the Wasserstein distance also decays exponentially and converges to the Wasserstein distance of the degree distribution. In addition, the expected Wasserstein distance is directly connected to the generalization error through our proposed bound, highlighting its practical significance. This connection further explains why oversmoothing measures show an ambiguous relationship with performance, with recent study even suggesting that no clear correlation has been observed (Heo et al., 2025). The key limitation of those measures is that they ignore class information, whereas our bound explicitly shows that inter-class feature distances should increase while intra-class distances should decrease. Hence, enforcing conventional oversmoothing measures to remain high may not be a proper strategy.

The second line of work (Wu et al., 2023) analyzes the effect of depth by decomposing it into two phenomena: the reduction of inter-class mean differences, termed the "mixing effect," and the reduction of intra-class variance, termed the "denoising effect." They seek to identify the optimal depth by balancing these two effects, and their formulation is conceptually related to our analysis of feature concentration and separation. Nevertheless, their framework expresses the combined influence of these effects in terms of the Bayes error of an idealized classifier. This choice introduces two limitations: (i) the assumption of an ideal classifier departs from practical settings, and (ii) node features in graphs are not i.i.d., so the Bayes error may be meaningful for a single sampled feature but not for the error rate over the entire dataset. In contrast, our generalization error does not rely on the existence of an ideal classifier and is derived under the transductive setting, thereby providing a measure that explains the performance of GNNs across the entire dataset and overcomes the limitations of Bayes error.

## 7 CONCLUSION

In this work, we established new generalization bounds for graph neural networks in the transductive node classification setting. By formulating the bounds in terms of the 1-Wasserstein distance between training and test feature distributions, we provided a principled framework that avoids surrogate independence assumptions and directly leverages access to unlabeled test features. Our analysis yields both global and class-wise error bounds, showing how feature concentration, separation, and classifier margins jointly determine generalization. Specializing to SGC, we derived spectral characterizations that reveal how graph topology and propagation depth control the evolution of feature distributions.

**Ethical Considerations** This work is primarily theoretical and does not involve new datasets or human subjects. Nevertheless, our results pertain to graph neural networks, which are often applied to sensitive relational data such as social or biological networks. Stronger generalization guarantees may encourage broader deployment of GNNs in such domains, raising concerns about privacy, fairness, and potential misuse. We emphasize that our theoretical bounds should not be interpreted as guarantees of equitable or unbiased performance, and their responsible application requires careful consideration of dataset biases, privacy risks, and broader societal impacts.

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

## A PROOFS

### A.1 PROOF OF THE GLOBAL WASSERSTEIN BOUND

**Theorem 4.1** (Global Wasserstein bound in the transductive setting). *Let $\gamma > 0$. For any random split $\pi$, and all $f \circ \phi \in F \circ \Phi$,*

$$R_u(f \circ \phi; \pi) \leq R_{m,\gamma}(f \circ \phi; \pi) + \frac{M(f, \phi)}{\gamma} \mathcal{W}_1\left(\phi_{\#}\mu_{\mathcal{I}_{\text{train}}^{(\pi)}}, \phi_{\#}\mu_{\mathcal{I}_{\text{test}}^{(\pi)}}\right), \tag{1}$$

*where*

$$M(f, \phi) := \max_{i,j,y} \frac{|\rho_f(\phi(\mathbf{x}_i), y_i) - \rho_f(\phi(\mathbf{x}_j), y)|}{\|\phi(\mathbf{x}_i) - \phi(\mathbf{x}_j)\|} \quad \text{for } i \in \mathcal{I}_{\text{train}}^{(\pi)}, \ j \in \mathcal{I}_{\text{test}}^{(\pi)}, \ y \in \mathcal{Y}.$$

*Proof.* The goal of the theorem is bounding the generalization gap between the zero-one loss of test set $R_u(f \circ \phi; \pi)$ and the $\gamma$-margin loss of train set $R_{m,\gamma}(f \circ \phi; \pi)$ of a model $f \circ \phi$ and permutation $\pi$, where:

$$R_u(f \circ \phi; \pi) := \frac{1}{u} \sum_{i \in \mathcal{I}_{\text{test}}^{(\pi)}} \mathbb{1}_{\rho_f(\phi(\mathbf{x}_i; \mathbf{X}, \mathbf{A}), y_i) \leq 0}, \tag{4}$$

and

$$R_{m,\gamma}(f \circ \phi; \pi) := \frac{1}{m} \sum_{i \in \mathcal{I}_{\text{train}}^{(\pi)}} \mathbb{1}_{\rho_f(\phi(\mathbf{x}_i; \mathbf{X}, \mathbf{A}), y_i) \leq \gamma}, \tag{5}$$

with $\gamma > 0$. We introduce a margin loss $L_\gamma$ with $\gamma > 0$ defined by $L_\gamma(u) := \mathbb{1}_{u \leq 0} + (1 - \frac{u}{\gamma})\mathbb{1}_{0 \leq u \leq \gamma}$. We first derive the upper bound on the difference of the margin loss on the test and train sets, i.e.:

$$\frac{1}{u} \sum_{i \in \mathcal{I}_{\text{test}}^{(\pi)}} L_\gamma(\rho_f(\phi(\mathbf{x}_i), y_i)) - \frac{1}{m} \sum_{j \in \mathcal{I}_{\text{train}}^{(\pi)}} L_\gamma(\rho_f(\phi(\mathbf{x}_j), y_j)). \tag{6}$$

By using the empirical distribution $\mu_{\mathcal{I}} := \frac{1}{|\mathcal{I}|} \sum_{i \in \mathcal{I}} \delta(\mathbf{x}_i)$, where $\delta(\cdot)$ denotes the Dirac delta function, the empirical mean can be represented as the expectation with respect to the empirical distribution. This allows us to rewrite Equation (6) as:

$$\mathbb{E}_{\mathbf{x} \sim \mu_{\mathcal{I}_{\text{test}}^{(\pi)}}}[L_\gamma(\rho_f(\phi(\mathbf{x}), y_{\mathbf{x}}))] - \mathbb{E}_{\mathbf{x}' \sim \mu_{\mathcal{I}_{\text{train}}^{(\pi)}}}[L_\gamma(\rho_f(\phi(\mathbf{x}'), y_{\mathbf{x}'}))]$$

$$= \int L_\gamma(\rho_f(\phi(\mathbf{x}), y_{\mathbf{x}}))d\mu_{\mathcal{I}_{\text{test}}^{(\pi)}} - \int L_\gamma(\rho_f(\phi(\mathbf{x}'), y_{\mathbf{x}'}))d\mu_{\mathcal{I}_{\text{train}}^{(\pi)}}$$

$$= \int \left(L_\gamma(\rho_f(\phi(\mathbf{x}), y_{\mathbf{x}})) - L_\gamma(\rho_f(\phi(\mathbf{x}'), y_{\mathbf{x}'}))\right)d\,T(\mathbf{x}, \mathbf{x}')$$

$$\leq \int \left\|L_\gamma(\rho_f(\phi(\mathbf{x}), y_{\mathbf{x}})) - L_\gamma(\rho_f(\phi(\mathbf{x}'), y_{\mathbf{x}'}))\right\|d\,T(\mathbf{x}, \mathbf{x}')$$

$$\leq \frac{1}{\gamma} \int \|\rho_f(\phi(\mathbf{x}), y_{\mathbf{x}}) - \rho_f(\phi(\mathbf{x}'), y_{\mathbf{x}'})\|d\,T(\mathbf{x}, \mathbf{x}') \tag{7}$$

where $y_{\mathbf{x}}$ and $y_{\mathbf{x}'}$ denote the labels of $\mathbf{x}$ and $\mathbf{x}'$, respectively. The last inequality in Equation (7) is based on the fact that $L_\gamma$ is $\frac{1}{\gamma}$-Lipschitz.

Now, we will bound $\|\rho_f(\phi(\mathbf{x}), y_{\mathbf{x}}) - \rho_f(\phi(\mathbf{x}'), y_{\mathbf{x}'})\|$ from our samples using an advantage of transductive settings. Since we cannot access the label of a test sample ($y_{\mathbf{x}}$), we define $M$ with all samples $\mathbf{x}, \mathbf{x}' \in \{\mathbf{x}_i\}_{i=1}^{m+u}$ and $y, y' \in \mathcal{Y}$. Define:

$$M(f, \phi) := \max_{\mathbf{x}, \mathbf{x}', y'} \frac{\|\rho_f(\phi(\mathbf{x}), y_{\mathbf{x}}) - \rho_f(\phi(\mathbf{x}'), y')\|}{\|\phi(\mathbf{x}) - \phi(\mathbf{x}')\|} \quad \text{for } \mathbf{x} \neq \mathbf{x}' \tag{8}$$

Then,

$$\|\rho_f(\phi(\mathbf{x}), y_\mathbf{x}) - \rho_f(\phi(\mathbf{x}'), y_{\mathbf{x}'})\| \leq M(f, \phi) \|\phi(\mathbf{x}) - \phi(\mathbf{x}')\| \tag{9}$$

From the Equation (7) and Equation (9), we have:

$$\mathbb{E}_{\mathbf{x} \sim \mu_{\mathcal{I}_{\text{test}}^{(\pi)}}}[L_\gamma(\rho_f(\phi(\mathbf{x}), y))] - \mathbb{E}_{\mathbf{x}' \sim \mu_{\mathcal{I}_{\text{train}}^{(\pi)}}}[L_\gamma(\rho_f(\phi(\mathbf{x}'), y'))]$$

$$\leq \int \Big( L_\gamma\big(\rho_f(\phi(\mathbf{x}), y)\big) - L_\gamma\big(\rho_f(\phi(\mathbf{x}'), y')\big)\Big) dT(\mathbf{x}, \mathbf{x}')$$

$$\leq \frac{1}{\gamma} \int \|\rho_f(\phi(\mathbf{x}), y) - \rho_f(\phi(\mathbf{x}'), y')\| dT(\mathbf{x}, \mathbf{x}')$$

$$\leq \frac{M(f, \phi)}{\gamma} \int \|\phi(\mathbf{x}) - \phi(\mathbf{x}')\| dT(\mathbf{x}, \mathbf{x}') \tag{10}$$

Since Equation (10) holds for any couplings $T \in \mathcal{H}(\mu_{\mathcal{I}_{\text{train}}^{(\pi)}}, \mu_{\mathcal{I}_{\text{test}}^{(\pi)}})$, we have:

$$\mathbb{E}_{\mathbf{x} \sim \mu_{\mathcal{I}_{\text{test}}^{(\pi)}}}[L_\gamma(\rho_f(\phi(\mathbf{x}), y))] - \mathbb{E}_{\mathbf{x}' \sim \mu_{\mathcal{I}_{\text{train}}^{(\pi)}}}[L_\gamma(\rho_f(\phi(\mathbf{x}'), y'))]$$

$$\leq \inf_{T \in \mathcal{U}} \frac{M(f, \phi)}{\gamma} \int_{T \in \mathcal{U}} \|\phi(\mathbf{x}) - \phi(\mathbf{x}')\| dT(\mathbf{x}, \mathbf{x}')$$

$$= \frac{M(f, \phi)}{\gamma} \mathcal{W}(\phi_{\#\mu_{\mathcal{I}_{\text{test}}}}, \phi_{\#\mu_{\mathcal{I}_{\text{test}}}}) \tag{11}$$

Putting together Equation (6) and Equation (11), then

$$\frac{1}{u} \sum_{i \in \mathcal{I}_{\text{test}}^{(\pi)}} L_\gamma(\rho_f(\phi(\mathbf{x}_i), y_i)) \leq \frac{1}{m} \sum_{i \in \mathcal{I}_{\text{train}}^{(\pi)}} L_\gamma(\rho_f(\phi(\mathbf{x}_j), y_j)) + \frac{M(f, \phi)}{\gamma} \mathcal{W}_1\big(\phi_{\#\mu_{\mathcal{I}_{\text{test}}^{(\pi)}}}, \phi_{\#\mu_{\mathcal{I}_{\text{train}}^{(\pi)}}}\big) .$$

Since $\mathbb{1}_{u \leq 0} \leq L_\gamma(u) \leq \mathbb{1}_{u \leq \gamma}$ for all $u$, we have:

$$R_u(f \circ \phi; \pi) \leq \frac{1}{u} \sum_{i \in \mathcal{I}_{\text{test}}^{(\pi)}} L_\gamma(\rho_f(\phi(\mathbf{x}_i), y_i))$$

$$\leq \frac{1}{m} \sum_{i \in \mathcal{I}_{\text{train}}^{(\pi)}} L_\gamma(\rho_f(\phi(\mathbf{x}_j), y_j)) + \frac{M(f, \phi)}{\gamma} \mathcal{W}_1\big(\phi_{\#\mu_{\mathcal{I}_{\text{test}}^{(\pi)}}}, \phi_{\#\mu_{\mathcal{I}_{\text{train}}^{(\pi)}}}\big)$$

$$\leq R_{m, \gamma}(f \circ \phi; \pi) + \frac{M(f, \phi)}{\gamma} \mathcal{W}_1\big(\phi_{\#\mu_{\mathcal{I}_{\text{test}}^{(\pi)}}}, \phi_{\#\mu_{\mathcal{I}_{\text{train}}^{(\pi)}}}\big) ,$$

which completes the proof. $\qquad\square$

## A.2 PROOF OF THE CLASS-WISE WASSERSTEIN BOUND

**Theorem 4.2** (Class-wise Wasserstein bound in the transductive setting). *Let $\gamma > 0$. Then, with probability at least $1 - \delta$ over the random split $\pi$, for all $f \circ \phi \in F \circ \Phi$,*

$$R_u(f \circ \phi; \pi) \leq R_{m,\gamma}(f \circ \phi; \pi) + \mathbb{E}_{\pi'}\left[\sum_{c=1}^{K}\left|\frac{u_c^{(\pi')}}{u} - \frac{m_c^{(\pi')}}{m}\right|\right]$$

$$+ \sum_{c=1}^{K}\frac{M_c(f,\phi)}{\gamma}\mathbb{E}_{\pi'}\left[\frac{m_c^{(\pi')}}{m}\,\mathcal{W}_1\left(\phi_{\#}\mu_{\mathcal{I}_{\text{train,c}}^{(\pi')}}, \phi_{\#}\mu_{\mathcal{I}_{\text{test,c}}^{(\pi')}}\right)\right] + \varepsilon_\delta, \quad (2)$$

*where*

$$M_c(f,\phi) := \max_{i,j}\frac{|\rho_f(\phi(\mathbf{x}_i),c) - \rho_f(\phi(\mathbf{x}_j),c)|}{\|\phi(\mathbf{x}_i) - \phi(\mathbf{x}_j)\|} \quad \text{for } i \neq j \text{ and } i,j \in \mathcal{I}_{\text{train,c}}^{(\pi)} \cup \mathcal{I}_{\text{test}}^{(\pi)},$$

$$\varepsilon_\delta = \sqrt{\frac{m\,u\,\beta^2}{2\,(m+u-\frac{1}{2})}\left(1 - \frac{1}{2\max\{m,u\}}\right)^{-1}\ln\frac{1}{\delta}}, \quad \text{and} \quad \beta = \frac{1}{m} + \frac{1}{u}.$$

*Proof.* The goal of the theorem is bounding the generalization gap between the zero-one loss of test set $R_u(f \circ \phi; \pi)$ and the $\gamma$-margin loss of train set $R_{m,\gamma}(f \circ \phi; \pi)$ of a model $f \circ \phi$ and permutation $\pi$, where:

$$R_u(f \circ \phi; \pi) := \frac{1}{u}\sum_{i \in \mathcal{I}_{\text{test}}^{(\pi)}}\mathbb{1}_{\rho_f(\phi(\mathbf{x}_i;\mathbf{X},\mathbf{A}),y_i)\leq 0}, \quad (12)$$

and

$$R_{m,\gamma}(f \circ \phi; \pi) := \frac{1}{m}\sum_{i \in \mathcal{I}_{\text{train}}^{(\pi)}}\mathbb{1}_{\rho_f(\phi(\mathbf{x}_i;\mathbf{X},\mathbf{A}),y_i)\leq \gamma}, \quad (13)$$

with $\gamma > 0$. We introduce a margin loss $L_\gamma$ with $\gamma > 0$ defined by $L_\gamma(u) = \mathbb{1}_{u\leq 0} + (1 - \frac{u}{\gamma})\mathbb{1}_{0<u\leq\gamma}$, which satisfies $\mathbb{1}_{u\leq 0} \leq L_\gamma(u) \leq \mathbb{1}_{u\leq\gamma}$ for all $u$. By simplifying $\ell_{\gamma,f}(z,y) = L_\gamma(\rho_f(z,y))$ and $\phi_{\mathcal{G}}(\cdot) = \phi(\cdot|\mathbf{X},\mathbf{A})$, we have:

$$R_u(f \circ \phi; \pi) \leq \frac{1}{u}\sum_{i \in \mathcal{I}_{\text{test}}^{(\pi)}}\ell_{\gamma,f}(\phi_{\mathcal{G}}(\mathbf{x}_i),y_i)$$

$$=\frac{1}{m}\sum_{j \in \mathcal{I}_{\text{train}}^{(\pi)}}\ell_{\gamma,f}(\phi_{\mathcal{G}}(\mathbf{x}_j),y_j) + \frac{1}{u}\sum_{i \in \mathcal{I}_{\text{test}}^{(\pi)}}\ell_{\gamma,f}(\phi_{\mathcal{G}}(\mathbf{x}_i),y_i) - \frac{1}{m}\sum_{j \in \mathcal{I}_{\text{train}}^{(\pi)}}\ell_{\gamma,f}(\phi_{\mathcal{G}}(\mathbf{x}_j),y_j)$$

$$\leq\frac{1}{m}\sum_{j \in \mathcal{I}_{\text{train}}^{(\pi)}}\ell_{\gamma,f}(\phi_{\mathcal{G}}(\mathbf{x}_j),y_j) + \sup_{f \in \mathcal{F}}\left(\frac{1}{u}\sum_{i \in \mathcal{I}_{\text{test}}^{(\pi)}}\ell_{\gamma,f}(\phi_{\mathcal{G}}(\mathbf{x}_i),y_i) - \frac{1}{m}\sum_{j \in \mathcal{I}_{\text{train}}^{(\pi)}}\ell_{\gamma,f}(\phi_{\mathcal{G}}(\mathbf{x}_j),y_j)\right)$$

$$\leq R_{m,\gamma}(f \circ \phi; \pi) + \sup_{f \in \mathcal{F}}\left(\frac{1}{u}\sum_{i \in \mathcal{I}_{\text{test}}^{(\pi)}}\ell_{\gamma,f}(\phi_{\mathcal{G}}(\mathbf{x}_i),y_i) - \frac{1}{m}\sum_{j \in \mathcal{I}_{\text{train}}^{(\pi)}}\ell_{\gamma,f}(\phi_{\mathcal{G}}(\mathbf{x}_j),y_j)\right). \quad (14)$$

Define

$$\Delta^{(\pi)}(f) := \frac{1}{u}\sum_{i \in \mathcal{I}_{\text{test}}^{(\pi)}}\ell_{\gamma,f}(\phi_{\mathcal{G}}(\mathbf{x}_i),y_i) - \frac{1}{m}\sum_{i \in \mathcal{I}_{\text{train}}^{(\pi)}}\ell_{\gamma,f}(\phi_{\mathcal{G}}(\mathbf{x}_i),y_i) \quad (15)$$

Then $\Delta^{(\pi)}(f)$ satisfies the assumption to apply the concentration inequality provided by El-Yaniv & Pechony (2009) (See the Definition A.2 and Lemma A.3 for details), we have with probability

at least $1 - \delta$,

$$\sup_{f \in \mathcal{F}} \left[ \Delta^{(\pi)}(f) \right] \leq \mathbb{E}_{\pi'} \sup_{f \in \mathcal{F}} \left[ \Delta^{(\pi')}(f) \right] + \sqrt{ \frac{m \, u \, (\frac{1}{m} + \frac{1}{u})^2}{2 \, (m + u - \frac{1}{2})} \left( 1 - \frac{1}{2 \max\{m, u\}} \right)^{-1} \ln \frac{1}{\delta} } \; . \quad (16)$$

To better understand the role of class-wise Wasserstein distance, we will analyze the generalization error for each class individually. By decomposing the $\Delta^{(\pi')}(f)$ into classes and simplifying $m_c^{(\pi)} = |\mathcal{I}_{\text{train},c}^{(\pi)}|$ and $u_c^{(\pi)} = |\mathcal{I}_{\text{test},c}^{(\pi)}|$, we get:

$$\sup_{f \in \mathcal{F}} \left[ \Delta^{(\pi')}(f) \right] = \sup_{f \in \mathcal{F}} \left[ \frac{1}{u} \sum_{i \in \mathcal{I}_{\text{test}}^{(\pi')}} \ell_{\gamma,f}(\phi_{\mathcal{G}}(\mathbf{x}_i), y_i) - \frac{1}{m} \sum_{i \in \mathcal{I}_{\text{train}}^{(\pi')}} \ell_{\gamma,f}(\phi_{\mathcal{G}}(\mathbf{x}_i), y_i) \right]$$

$$= \sup_{f \in \mathcal{F}} \left[ \sum_{c=1}^{K} \frac{m_c^{(\pi')}}{m} \underbrace{\left( \frac{1}{u_c^{(\pi')}} \sum_{i \in \mathcal{I}_{\text{test},c}^{(\pi')}} \ell_{\gamma,f}(\phi_{\mathcal{G}}(\mathbf{x}_i), c) - \frac{1}{m_c^{(\pi')}} \sum_{i \in \mathcal{I}_{\text{train},c}^{(\pi')}} \ell_{\gamma,f}(\phi_{\mathcal{G}}(\mathbf{x}_i), c) \right)}_{:= \Delta_c^{(\pi')}(f)} \right.$$

$$\left. + \sum_{c=1}^{K} \left( \frac{u_c^{(\pi')}}{u} - \frac{m_c^{(\pi')}}{m} \right) \left( \frac{1}{u_c^{(\pi')}} \sum_{i \in \mathcal{I}_{\text{test},c}^{(\pi')}} \ell_{\gamma,f}(\phi_{\mathcal{G}}(\mathbf{x}_i), c) \right) \right]$$

$$\leq \sup_{f \in \mathcal{F}} \left( \sum_{c=1}^{K} \Delta_c^{(\pi')}(f) \right) + \sup_{f \in \mathcal{F}} \left[ \sum_{c=1}^{K} \left( \frac{u_c^{(\pi')}}{u} - \frac{m_c^{(\pi')}}{m} \right) \left( \frac{1}{u_c^{(\pi')}} \sum_{i \in \mathcal{I}_{\text{test},c}^{(\pi')}} \ell_{\gamma,f}(\phi_{\mathcal{G}}(\mathbf{x}_i), c) \right) \right]$$

$$\leq \sup_{f \in \mathcal{F}} \left( \sum_{c=1}^{K} \Delta_c^{(\pi')}(f) \right) + \sup_{f \in \mathcal{F}} \left[ \sum_{c=1}^{K} \left| \frac{u_c^{(\pi')}}{u} - \frac{m_c^{(\pi')}}{m} \right| \left( \frac{1}{u_c^{(\pi')}} \sum_{i \in \mathcal{I}_{\text{test},c}^{(\pi')}} \ell_{\gamma,f}(\phi_{\mathcal{G}}(\mathbf{x}_i), c) \right) \right]$$

$$\leq \sup_{f \in \mathcal{F}} \left( \sum_{c=1}^{K} \Delta_c^{(\pi')}(f) \right) + \sum_{c=1}^{K} \left| \frac{u_c^{(\pi')}}{u} - \frac{m_c^{(\pi')}}{m} \right| \quad (17)$$

Then, we have:

$$\mathbb{E}_{\pi'} \sup_{f \in \mathcal{F}} \left[ \Delta^{(\pi')}(f) \right] \leq \mathbb{E}_{\pi'} \left[ \sup_{f \in \mathcal{F}} \sum_{c=1}^{K} \left( \Delta_c^{(\pi')}(f) \right) + \sum_{c=1}^{K} \left| \frac{u_c^{(\pi')}}{u} - \frac{m_c^{(\pi')}}{m} \right| \right]$$

$$\leq \sum_{c=1}^{K} \mathbb{E}_{\pi'} \left[ \sup_{f \in \mathcal{F}} \left( \Delta_c^{(\pi')}(f) \right) \right] + \sum_{c=1}^{K} \mathbb{E}_{\pi'} \left[ \left| \frac{u_c^{(\pi')}}{u} - \frac{m_c^{(\pi')}}{m} \right| \right] \quad (18)$$

Define:

$$M_c(f, \phi) := \max_{i,j} \frac{|\rho_f(\phi(\mathbf{x}_i), c) - \rho_f(\phi(\mathbf{x}_j), c)|}{\|\phi(\mathbf{x}_i) - \phi(\mathbf{x}_j)\|} \quad \text{for } i \neq j \text{ and } i, j \in \mathcal{I}_{\text{train},c}^{(\pi)} \cup \mathcal{I}_{\text{test}}^{(\pi)},$$

Since $L_\gamma$ is $\frac{1}{\gamma}$-Lipschitz and $|\rho_f(\phi(\mathbf{x}_i), c) - \rho_f(\phi(\mathbf{x}_j), c)| \leq M_c(f, \phi)\|\phi(\mathbf{x}_i) - \phi(\mathbf{x}_j)\|$, we can repeat the derivations in Equation (10). Then we have:

$$\sup_{f \in \mathcal{F}} [\Delta_c^{(\pi')}(f)] = \sup_{f \in \mathcal{F}} \left[ \frac{m_c^{(\pi')}}{m} \left( \mathbb{E}_{\mathbf{x} \sim \mu_{\mathcal{I}_{\text{test},c}^{(\pi')}}} \ell_{\gamma,f}(\phi_{\mathcal{G}}(\mathbf{x}), c) - \mathbb{E}_{\mathbf{x} \sim \mu_{\mathcal{I}_{\text{train},c}^{(\pi')}}} \ell_{\gamma,f}(\phi_{\mathcal{G}}(\mathbf{x}), c) \right) \right]$$

$$\leq \frac{m_c^{(\pi)}}{m} \frac{M_c(f, \phi)}{\gamma} \mathcal{W}_1 \left( \phi_\# \mu_{\mathcal{I}_{\text{test},c}^{(\pi')}}, \phi_\# \mu_{\mathcal{I}_{\text{train},c}^{(\pi')}} \right) \; . \quad (19)$$

Putting together Equations (16) to (19), with probability at least $1 - \delta$, we have:

$$
\sup_{f \in \mathcal{F}} \left[ \frac{1}{u} \sum_{i \in \mathcal{I}_{\text{test}}^{(\pi)}} \ell_{\gamma,f}(\phi_{\mathcal{G}}(\mathbf{x}_i), y_i) - \frac{1}{m} \sum_{j \in \mathcal{I}_{\text{train}}^{(\pi)}} \ell_{\gamma,f}(\phi_{\mathcal{G}}(\mathbf{x}_j), y_j) \right]
$$

$$
\leq \sum_{c=1}^{K} \mathbb{E}_{\pi'} \frac{m_c^{(\pi)}}{m} \frac{M_c(f, \phi)}{\gamma} \mathcal{W}_1(\phi_{\#}\mu_{\mathcal{I}_{\text{test},c}^{(\pi)}}, \phi_{\#}\mu_{\mathcal{I}_{\text{train},c}^{(\pi)}}) + \mathbb{E}_{\pi'} \left[ \sum_{c=1}^{K} \left| \frac{u_c^{(\pi')}}{u} - \frac{m_c^{(\pi')}}{m} \right| \right] + \varepsilon_\delta \,,
$$
$$
\tag{20}
$$

where $\varepsilon_\delta = \sqrt{\frac{m\,u\,(\frac{1}{m} + \frac{1}{u})^2}{2\,(m+u-\frac{1}{2})} \left( 1 - \frac{1}{2\max\{m,u\}} \right)^{-1} \ln \frac{1}{\delta}}$. Plugging Equation (20) into Equation (14),

we obtain the bound stated in the theorem, which completes the proof. $\qquad\square$

---

**Lemma A.1.** *Let $f = [f_1, \ldots, f_K] \in \mathcal{F} : \mathcal{Z} \to \mathbb{R}^K$ with each component $f_j : \mathcal{Z} \to \mathbb{R}$ being $L_j$-Lipschitz, i.e., $\|f_j(z) - f_j(z')\| \leq L_j \|z - z'\|$ for all $z, z' \in \mathcal{Z}$. For any fixed $y \in \{1, \ldots, K\}$, the margin $\rho_f(z, y) := f_y(z) - \max_{y' \neq y} f_{y'}(z)$ is Lipschitz in its first argument.*

---

**Definition A.2** $((m, u)$-permutation symmetry (El-Yaniv & Pechyony, 2009)$)$**.** Let $[m + u] = \{1, \ldots, m, m+1, \ldots, m+u\}$ and $[m+1, m+u] = \{m+1, \ldots, m+u\}$, and $S_{m+u}$ be the set of permutation functions on $[m + u]$. Define the block-preserving permutation function.

$$
H_{m,u} = \big\{ \sigma \in S_{m+u} : \sigma([m]) = [m] \text{ and } \sigma([m+1, m+u]) = [m+1, m+u] \big\}.
$$

A function $g : S_{m+u} \to \mathbb{R}$ is called $(m, u)$-*permutation symmetric* if

$$
g(\pi) = g(\sigma \circ \pi) \qquad \text{for all } \pi \in S_{m+u} \text{ and all } \sigma \in H_{m,u}.
$$

Equivalently, $g(\pi)$ depends only on the unordered split $\big(\pi([m]), \pi([m+1, m+u])\big)$, i.e., the training/test partition, and not on the ordering within each block.

---

**Lemma A.3** (Concentration Inequality on Transductive Setting (El-Yaniv & Pechyony, 2009))**.** *Let $S_{m+u}$ be the set of permutation functions on $[m + u] = \{1, \ldots, m + u\}$. Let $\pi \in S_{m+u}$ and $g : S_{m+u} \to \mathbb{R}$ be $(m, u)$-permutation symmetric. For $i \in \{1, \ldots, m\}$ and $j \in \{m+1, \ldots, m+u\}$, let $\tau_{ij} \in S_{m+u}$ be the transposition of positions $i$ and $j$, and write $\pi^{(ij)} := \tau_{ij} \circ \pi$. If for some $\beta > 0$,*

$$
\big| g(\pi) - g(\pi^{(ij)}) \big| \leq \beta \qquad \text{for all } i \leq m < j,
$$

*then for every $\varepsilon > 0$,*

$$
\Pr\big\{ g(\pi) - \mathbb{E}_{\pi'}[g(\pi')] \geq \varepsilon \big\} \leq \exp\left( -\frac{2\,\varepsilon^2\,(m + u - \frac{1}{2})}{m\,u\,\beta^2 \left( 1 - \frac{1}{2\max\{m,u\}} \right)} \right),
$$

*where $\pi' \in S_{m+u}$ is an independent to $\pi$. Equivalently, with probability at least $1 - \delta$,*

$$
g(\pi) \leq \mathbb{E}_{\pi'}[g(\pi')] + \varepsilon_\delta, \qquad \varepsilon_\delta = \sqrt{\frac{m\,u\,\beta^2}{2\,(m + u - \frac{1}{2})} \left( 1 - \frac{1}{2\max\{m,u\}} \right)^{-1} \ln \frac{1}{\delta}}.
$$

### A.3 PROOF OF THE WASSERSTEIN DISTANCE ON SGC

**Theorem 6.1** (Wasserstein distance of encoded features of SGC). *Let $\mathcal{S}, \mathcal{T} \subseteq [N]$ be any nonempty index subsets. Then, for every propagation depth $\ell \in \mathbb{N}$,*

$$\mathcal{W}_1\big(\phi_\#^{(\ell)}\mu_\mathcal{S}, \phi_\#^{(\ell)}\mu_\mathcal{T}\big) \le C_1\big(\mathbf{X}, \hat{\mathbf{A}}\big)\, \mathcal{W}_1\big(d_\#\mu_\mathcal{S},\, d_\#\mu_\mathcal{T}\big) + C_2(\mathbf{X})\, \rho_\perp(\hat{\mathbf{A}})^\ell, \qquad (3)$$

*where $\rho_\perp(\hat{\mathbf{A}}) := \max_{k \in \{2,\dots,N\}}\{|\lambda_k(\hat{\mathbf{A}})| : \lambda_1(\hat{\mathbf{A}}) = 1\} \in [0,1)$ is the nontrivial spectral radius, and $C_1(\mathbf{X}, \hat{\mathbf{A}})$, $C_2(\mathbf{X})$ are finite constants depending only on $(\mathbf{X}, \hat{\mathbf{A}})$.*

*Proof.* For $i \in \mathcal{I}_\mathcal{S}$ and $j \in \mathcal{I}_\mathcal{U}$, write the encoded rows with standard $i$- and $j$-th basis vector

$$\mathbf{z}_i := (\hat{\mathbf{A}}^\ell \mathbf{X})_{i\cdot} = \mathbf{e}_i^\top \hat{\mathbf{A}}^\ell \mathbf{X}, \qquad \mathbf{w}_j := (\hat{\mathbf{A}}^\ell \mathbf{X})_{j\cdot} = \mathbf{e}_j^\top \hat{\mathbf{A}}^\ell \mathbf{X}.$$

For each class $c' \in \{1, \dots, K\}$, let the selector $\mathbf{S}_{c'} \in \{0,1\}^{N \times N}$ be the diagonal matrix $(\mathbf{S}_{c'})_{vv} := \mathbf{1}\{y_v = c'\}$. This keeps only rows of class $c'$ and zeros out the rest. Class-wise message decomposition gives

$$\mathbf{z}_i = \sum_{c'=1}^K \mathbf{e}_i^\top \hat{\mathbf{A}}^\ell \mathbf{S}_{c'}\, \mathbf{X}, \qquad \mathbf{w}_j = \sum_{c'=1}^K \mathbf{e}_j^\top \hat{\mathbf{A}}^\ell \mathbf{S}_{c'}\, \mathbf{X},$$

hence

$$\mathbf{z}_i - \mathbf{w}_j = \sum_{c'=1}^K \big(\mathbf{e}_i^\top \hat{\mathbf{A}}^\ell \mathbf{S}_{c'} - \mathbf{e}_j^\top \hat{\mathbf{A}}^\ell \mathbf{S}_{c'}\big)\, \mathbf{X}.$$

Since $\|\mathbf{A}\mathbf{B}\| \le \|\mathbf{A}\|_2 \|\mathbf{B}\|$,

$$\|\mathbf{z}_i - \mathbf{w}_j\| \le \sum_{c'=1}^K \big\|\big(\mathbf{e}_i^\top \hat{\mathbf{A}}^\ell \mathbf{S}_{c'} - \mathbf{e}_j^\top \hat{\mathbf{A}}^\ell \mathbf{S}_{c'}\big)\mathbf{X}\big\| \le \sum_{c'=1}^K \|\mathbf{X}_{c'}\|_2\, \big\|(\mathbf{e}_i - \mathbf{e}_j)^\top \hat{\mathbf{A}}^\ell \mathbf{S}_{c'}\big\|, \qquad (21)$$

where $\mathbf{X}_{c'}$ stacks $\{\mathbf{x}_v\}_{y_v = c'}$ and $\|\mathbf{X}_{c'}\|_2 \le \sqrt{|\mathcal{I}_{c'}|}\, \max_{y_v = c'} \|\mathbf{x}_v\|$.

Let $\mathbf{u}_1 := \tilde{\mathbf{D}}^{1/2}\mathbf{1}$. $\mathbf{u}_1$ is an eigenvector of $\hat{\mathbf{A}}$ with an eigenvalue 1, since

$$\hat{\mathbf{A}}\,\mathbf{u}_1 = \tilde{\mathbf{D}}^{-1/2}\tilde{\mathbf{A}}\,\tilde{\mathbf{D}}^{-1/2}\tilde{\mathbf{D}}^{1/2}\mathbf{1} = \tilde{\mathbf{D}}^{-1/2}\tilde{\mathbf{A}}\,\mathbf{1} = \tilde{\mathbf{D}}^{-1/2}\tilde{\mathbf{D}}\,\mathbf{1} = \tilde{\mathbf{D}}^{1/2}\mathbf{1} = \mathbf{u}_1.$$

Define

$$\mathcal{U} := \mathrm{span}\{\mathbf{u}_1\}, \qquad \mathcal{U}^\perp := \{\mathbf{x} \in \mathbb{R}^N : \langle \mathbf{x}, \mathbf{u}_1 \rangle = 0\}.$$

The orthogonal projectors onto $\mathcal{U}$ and $\mathcal{U}^\perp$ are

$$\mathcal{P}_\mathcal{U} := \frac{\mathbf{u}_1 \mathbf{u}_1^\top}{\|\mathbf{u}_1\|^2}, \qquad \mathcal{P}_{\mathcal{U}^\perp} := \mathcal{I} - \mathcal{P}_\mathcal{U}, \qquad \|\mathbf{u}_1\|^2 = \sum_{v=1}^N \tilde{d}_v.$$

By the spectral theorem, there exists an orthonormal basis $\mathbf{Q} = [\mathbf{v}_1, \dots, \mathbf{v}_N]$ with $\hat{\mathbf{A}} = \mathbf{Q}\,\mathrm{diag}(1, \lambda_2, \dots, \lambda_N)\,\mathbf{Q}^\top$ with $1 = \lambda_1 \ge \lambda_2 \ge \cdots \ge \lambda_N > -1$ and we choose $\mathbf{v}_1 = \frac{\mathbf{u}_1}{\|\mathbf{u}_1\|}$. Then, for every $\ell \in \mathbb{N}$,

$$\hat{\mathbf{A}}^\ell = \mathbf{Q}\,\mathrm{diag}\big(1, \lambda_2^\ell, \dots, \lambda_N^\ell\big)\,\mathbf{Q}^\top = \underbrace{\mathbf{Q}\,\mathrm{diag}(1, 0, \dots, 0)\,\mathbf{Q}^\top}_{=\,\mathcal{P}_\mathcal{U}} + \underbrace{\mathbf{Q}\,\mathrm{diag}(0, \lambda_2^\ell, \dots, \lambda_N^\ell)\,\mathbf{Q}^\top}_{=\,\mathcal{P}_{\mathcal{U}^\perp}\,\hat{\mathbf{A}}^\ell\,\mathcal{P}_{\mathcal{U}^\perp}}.$$

Equivalently,

$$\hat{\mathbf{A}}^\ell = \mathcal{P}_\mathcal{U} + \mathcal{P}_{\mathcal{U}^\perp} \hat{\mathbf{A}}^\ell \mathcal{P}_{\mathcal{U}^\perp}$$

Moreover,

$$\left\| \mathcal{P}_{\mathcal{U}^\perp} \hat{\mathbf{A}}^\ell \mathcal{P}_{\mathcal{U}^\perp} \right\|_2 = \max_{k \geq 2} |\lambda_k(\hat{\mathbf{A}})|^\ell := \rho_\perp(\hat{\mathbf{A}})^\ell.$$

Therefore,

$$\left\| (\mathbf{e}_i - \mathbf{e}_j)^\top \hat{\mathbf{A}}^\ell \mathbf{S}_{c'} \right\| \leq \left\| (\mathbf{e}_i - \mathbf{e}_j)^\top \mathcal{P}_\mathcal{U} \mathbf{S}_{c'} \right\| + \left\| (\mathbf{e}_i - \mathbf{e}_j)^\top \mathcal{P}_{\mathcal{U}^\perp} \hat{\mathbf{A}}^\ell \mathcal{P}_{\mathcal{U}^\perp} \mathbf{S}_{c'} \right\|.$$

For the first term,

$$\left\| (\mathbf{e}_i - \mathbf{e}_j)^\top \mathcal{P}_\mathcal{U} \mathbf{S}_{c'} \right\| = \frac{|\sqrt{\tilde{d}_i} - \sqrt{\tilde{d}_j}|}{\sum_v \tilde{d}_v} \left\| \mathbf{S}_{c'} \mathbf{u}_1 \right\| = \frac{|\sqrt{\tilde{d}_i} - \sqrt{\tilde{d}_j}|}{\sum_v \tilde{d}_v} \sqrt{\sum_{y_v = c'} \tilde{d}_v}. \tag{22}$$

For the last term, using $\|\mathbf{e}_i - \mathbf{e}_j\| = \sqrt{2}$, $\|\mathcal{P}_{\mathcal{U}^\perp}\|_2 = \|\mathbf{S}_{c'}\|_2 = 1$ gives

$$
\begin{aligned}
\left\| (\mathbf{e}_i - \mathbf{e}_j)^\top \mathcal{P}_{\mathcal{U}^\perp} \hat{\mathbf{A}}^\ell \mathcal{P}_{\mathcal{U}^\perp} \mathbf{S}_{c'} \right\| &\leq \|\mathbf{e}_i - \mathbf{e}_j\| \left\| \mathcal{P}_{\mathcal{U}^\perp} \hat{\mathbf{A}}^\ell \mathcal{P}_{\mathcal{U}^\perp} \mathbf{S}_{c'} \right\|_2 \quad \text{(vector–matrix inequality)} \\
&\leq \|\mathbf{e}_i - \mathbf{e}_j\| \|\mathcal{P}_{\mathcal{U}^\perp}\|_2 \left\| \hat{\mathbf{A}}^\ell \mathcal{P}_{\mathcal{U}^\perp} \right\|_2 \|\mathbf{S}_{c'}\|_2 \quad \text{(submultiplicativity)} \\
&= \|\mathbf{e}_i - \mathbf{e}_j\| \left\| \hat{\mathbf{A}}^\ell \big|_{\mathcal{U}^\perp} \right\|_2 \quad (\|\mathcal{P}_{\mathcal{U}^\perp}\|_2 = \|\mathbf{S}_{c'}\|_2 = 1) \\
&= \|\mathbf{e}_i - \mathbf{e}_j\| \, \rho_\perp(\hat{\mathbf{A}})^\ell \quad \text{(spectral theorem on } \mathcal{U}^\perp) \\
&= \sqrt{2} \, \rho_\perp(\hat{\mathbf{A}})^\ell.
\end{aligned}
$$

Therefore,

$$\|\mathbf{z}_i - \mathbf{w}_j\| \leq \sum_{c'=1}^K \left( \sqrt{|\mathcal{I}_{c'}|} \max_{y_v = c'} \|\mathbf{x}_v\| \right) \left( \frac{\sqrt{\sum_{y_v=c'} \tilde{d}_v} |\sqrt{\tilde{d}_i} - \sqrt{\tilde{d}_j}|}{\sum_v \tilde{d}_v} + \sqrt{2} \, \rho_\perp(\hat{\mathbf{A}})^\ell \right). \tag{23}$$

Finally, plugging [Equation (23)](#) into the definition of the 1-Wasserstein distance,

$$
\begin{aligned}
\mathcal{W}_1\big(\phi_\# \mu_\mathcal{S}, \, \phi_\# \mu_\mathcal{T}\big) &\leq \min_\Gamma \sum_{i \in \mathcal{I}_{\text{train},c}} \sum_{j \in \mathcal{I}_{\text{test},c}} \Gamma_{ij} \|\mathbf{z}_i - \mathbf{w}_j\| \\
&\leq \min_\Gamma \sum_{i,j} \Gamma_{ij} \sum_{c'=1}^K \left( \sqrt{|\mathcal{I}_{c'}|} \max_{y_v = c'} \|\mathbf{x}_v\| \right) \\
&\qquad \left( \frac{\sqrt{\sum_{y_v=c'} \tilde{d}_v} |\sqrt{\tilde{d}_i} - \sqrt{\tilde{d}_j}|}{\sum_{v=1}^N \tilde{d}_v} + \sqrt{2} \, \rho_\perp(\hat{\mathbf{A}})^\ell \right) \\
&= \sum_{c'=1}^K \left( \sqrt{|\mathcal{I}_{c'}|} \max_{y_v = c'} \|\mathbf{x}_v\| \right) \\
&\qquad \left( \frac{\sqrt{\sum_{y_v=c'} \tilde{d}_v}}{\sum_{v=1}^N \tilde{d}_v} \min_\Gamma \sum_{i,j} \Gamma_{ij} |\sqrt{\tilde{d}_i} - \sqrt{\tilde{d}_j}| + \min_\Gamma \sum_{i,j} \Gamma_{ij} \sqrt{2} \, \rho_\perp(\hat{\mathbf{A}})^\ell \right).
\end{aligned}
$$

By the definition,

$$\min_\Gamma \sum_{i,j} \Gamma_{ij} |\sqrt{\tilde{d}_i} - \sqrt{\tilde{d}_j}| = \mathcal{W}_1\big(\mu_{\mathcal{I}_{\text{train},c}}^{\sqrt{d}}, \, \mu_{\mathcal{I}_{\text{test},c}}^{\sqrt{d}}\big),$$

where $\mu_{\mathcal{I}_{\text{train},c}}^{\sqrt{d}} := \frac{1}{|\mathcal{I}_{\text{train},c}|} \sum_{i \in \mathcal{I}_{\text{train},c}} \delta_{\sqrt{\tilde{d}_i}}$ and $\mu_{\mathcal{I}_{\text{test},c}}^{\sqrt{d}} := \frac{1}{|\mathcal{I}_{\text{test},c}|} \sum_{j \in \mathcal{I}_{\text{test},c}} \delta_{\sqrt{\tilde{d}_j}}$.

Therefore,

$$
\mathcal{W}_1\big(\phi_\# \mu_{\mathcal{I}_{\mathrm{train},c}},\, \phi_\# \mu_{\mathcal{I}_{\mathrm{test},c}}\big) \leq \bigg( \sum_{c'=1}^{K} \sqrt{|\mathcal{I}_{c'}|} \max_{y_v=c'} \|\mathbf{x}_v\| \frac{\sqrt{\sum_{y_v=c'} \tilde{d}_v}}{\sum_{v=1}^{N} \tilde{d}_v} \bigg) \mathcal{W}_1\big(\mu^{\sqrt{d}}_{\mathcal{I}_{\mathrm{train},c}},\, \mu^{\sqrt{d}}_{\mathcal{I}_{\mathrm{test},c}}\big)
$$

$$
+ \sum_{c'=1}^{K} \sqrt{|\mathcal{I}_{c'}|} \max_{y_v=c'} \|\mathbf{x}_v\| \sqrt{2}\, \rho_\perp(\hat{\mathbf{A}})^\ell .
$$

$\square$

## B    DATASET STATISTICS

We provide detailed statistics and explanations about the dataset used for the experiments in Table 2 and the paragraphs below.

Table 2: Statistics of the datasets utilized in the experiments.

| Dataset | # nodes | # edges | # features | # classes |
|---|---|---|---|---|
| Cora | 2,708 | 5,278 | 1,433 | 7 |
| CiteSeer | 3,327 | 4,552 | 3,703 | 6 |
| PubMed | 19,717 | 44,324 | 500 | 3 |
| Computers | 13,752 | 245,861 | 767 | 10 |
| Photo | 7,650 | 119,081 | 745 | 8 |
| Squirrel | 2,223 | 46,998 | 2,089 | 5 |
| Chameleon | 890 | 8,854 | 2,325 | 5 |
| Roman-empire | 22,662 | 32,927 | 300 | 18 |
| Amazon-ratings | 24,492 | 93,050 | 300 | 5 |

**Cora, CiteSeer, and PubMed**    Each node represents a paper, and an edge indicates a reference relationship between two papers. The task is to predict the research subjects of the papers.

**Computers and Photo**    Each node represents a product, and an edge indicates a high frequency of concurrent purchases of the two products. The task is to predict the product category.

**Squirrel and Chameleon**    Each node represents a Wikipedia page, and an edge indicates a link between two pages. The task is to predict the monthly traffic for each page. We use the classification version of the dataset, where labels are converted by dividing monthly traffic into five bins. We adopted the filtering process to prevent train-test data leakage as recommended by (Platonov et al., 2023).

**Roman-empire**    Each node represents a word extracted from the English Wikipedia article on the Roman Empire, and an edge indicates a grammatical or sequential relationship between words. The task is to predict the part-of-speech tag of each word.

**Amazon-ratings**    Each node represents a product from the Amazon co-purchasing network, and an edge indicates a frequent co-purchase between two products. The task is to predict the product category based on user co-purchasing patterns.

## C    FURTHER IMPLEMENTATION DETAILS

In this section, we explain how to calculate the expectation terms of Equation (2) in Theorem 4.2.

Table 3: Correlation values (with computation time in seconds) for estimating the expectation in Theorem 4.2 using different numbers of sampled permutations (1, 4, 16, 64). Results are reported for SGC and GCN on the Cora and Amazon-Ratings datasets under both the *oracle* and *approx* settings.

|  |  |  | 1 | 4 | 16 | 64 |
|---|---|---|---|---|---|---|
| SGC | Cora | oracle | 0.84 (0.50s) | 0.89 (1.99s) | 0.9 (8.22s) | 0.88 (23.7s) |
|  |  | approx | 0.84 (0.36s) | 0.87 (1.67s) | 0.87 (6.21s) | 0.86 (20.1s) |
|  | Amazon-ratings | oracle | 0.76 (9.69s) | 0.93 (39.8s) | 0.89 (147s) | 0.91 (611s) |
|  |  | approx | 0.76 (0.83s) | 0.91 (3.31s) | 0.87 (13.2s) | 0.92 (49.2s) |
| GCN | Cora | oracle | 0.7 (0.69s) | 0.81 (2.58s) | 0.75 (10.5s) | 0.78 (43.9s) |
|  |  | approx | 0.67 (0.17s) | 0.78 (0.68s) | 0.72 (2.84s) | 0.77 (12.1s) |
|  | Amazon-ratings | oracle | 0.9 (11.4s) | 0.91 (45.0s) | 0.88 (180s) | 0.94 (728s) |
|  |  | approx | 0.88 (0.72s) | 0.91 (2.89s) | 0.87 (11.5s) | 0.94 (46.8s) |

Each $\pi'$ is independently sampled from the uniform distribution over all $(m + u)!$ permutations. For a given permutation $\pi'$, we consider two estimators of the terms inside the expectations: *oracle* and *approx*.

The *oracle* approach assumes access to the labels of all nodes. Under this assumption, every quantity, i.e., $u_c^{(\pi')}$, $m_c^{(\pi')}$, and $\mathcal{W}_1\big(\phi_\#\mu_{\mathcal{I}_{\text{train,c}}^{(\pi')}}, \phi_\#\mu_{\mathcal{I}_{\text{test,c}}^{(\pi')}}\big)$, can be computed exactly.

In contrast, the *approx* approach only uses the labels of the training nodes, i.e., $\{y_i\}_{i \in \mathcal{I}_{\text{train}}^{(\pi)}}$, and approximates all terms. Specifically, *approx* replaces

$$u_c^{(\pi')} \approx |\mathcal{I}_{\text{test}}^{(\pi';\pi)}|, \quad m_c^{(\pi')} \approx |\mathcal{I}_{\text{train}}^{(\pi';\pi)}|,$$

$$\text{and} \quad \mathcal{W}_1\big(\phi_\#\mu_{\mathcal{I}_{\text{train,c}}^{(\pi')}}, \phi_\#\mu_{\mathcal{I}_{\text{test,c}}^{(\pi')}}\big) \approx \mathcal{W}_1\big(\phi_\#\mu_{\mathcal{I}_{\text{train,c}}^{(\pi';\pi)}}, \phi_\#\mu_{\mathcal{I}_{\text{test,c}}^{(\pi';\pi)}}\big),$$

where

$$\mathcal{I}_{\text{train,c}}^{(\pi';\pi)} = \mathcal{I}_{\text{train}}^{(\pi')} \cap \mathcal{I}_{\text{train,c}}^{(\pi)} \quad \text{and} \quad \mathcal{I}_{\text{test,c}}^{(\pi';\pi)} = \mathcal{I}_{\text{test}}^{(\pi')} \cap \mathcal{I}_{\text{train,c}}^{(\pi)}.$$

We then approximate each expectation over $\pi'$ by the empirical average over $T$ sampled permutations: $\mathbb{E}_{\pi'}[\cdot] \approx \frac{1}{T} \sum_{t=1}^{T}(\cdot)$.

We use $T = 4$, because larger sample sizes (16 or 64) do not yield consistent gains, whereas using smaller sizes (1) sometimes produces noticeably lower correlation. Based on this observation, we selected four permutations as the most efficient choice. The corresponding correlation results for SGC and GCN on Cora and Amazon-Ratings (with 1, 4, 16, and 64 samples) are reported in Table 3.

## D ADDITIONAL RESULTS

Table 4: Correlation between empirical error gap and generalization bounds across datasets and GNNs. *Global* reports our bound with $M(f, \phi)$, and *global*(0.9) reports with 0.9 percentile of $M(f, \phi)$ from Theorem 4.1, while *oracle* and *approx* correspond to the class-wise bound in Theorem 4.2 with and without test labels, respectively. Table shading reflects correlation: darker for lower, lighter for higher, indicating stronger alignment with performance. The *RC bound* (Esser et al., 2021) is shown for the applicable models.

| | | Cora | CiteSeer | PubMed | Computers | Photo | Squirrel | Chameleon | Roman-Empire | Amazon-Ratings |
|---|---|---|---|---|---|---|---|---|---|---|
| SGC | global | 0.01 | 0.26 | 0.16 | -0.42 | -0.07 | 0.03 | -0.51 | 0.59 | -0.07 |
| | global(0.9) | 0.92 | 0.96 | 0.96 | 0.19 | 0.84 | 0.95 | 0.76 | 0.80 | 0.98 |
| | oracle | 0.89 | 0.82 | 0.98 | 0.26 | 0.81 | 0.87 | 0.62 | 0.78 | 0.93 |
| | approx | 0.87 | 0.83 | 0.98 | 0.20 | 0.81 | 0.85 | 0.58 | 0.78 | 0.91 |
| | RC bound | -0.92 | -0.97 | -0.38 | -0.06 | -0.25 | -0.94 | -0.94 | -0.56 | -0.77 |
| GCN | global | 0.32 | 0.26 | 0.23 | 0.09 | 0.31 | 0.23 | -0.10 | 0.31 | 0.39 |
| | global(0.9) | 0.82 | 0.79 | 0.82 | 0.77 | 0.79 | 0.70 | 0.48 | 0.68 | 0.93 |
| | oracle | 0.81 | 0.66 | 0.69 | 0.61 | 0.63 | 0.42 | 0.06 | 0.69 | 0.91 |
| | approx | 0.78 | 0.60 | 0.68 | 0.56 | 0.57 | 0.38 | 0.01 | 0.65 | 0.91 |
| | RC bound | 0.29 | 0.48 | 0.35 | -0.01 | -0.27 | -0.75 | -0.82 | 0.35 | 0.20 |
| GCNII | global | 0.41 | 0.46 | 0.80 | -0.33 | -0.31 | -0.15 | -0.14 | 0.79 | 0.64 |
| | global(0.9) | 0.89 | 0.82 | 0.67 | 0.88 | 0.90 | 0.18 | -0.39 | 0.68 | 0.95 |
| | oracle | 0.82 | 0.77 | 0.66 | 0.83 | 0.87 | 0.17 | -0.28 | 0.70 | 0.91 |
| | approx | 0.77 | 0.75 | 0.67 | 0.81 | 0.84 | 0.17 | -0.31 | 0.70 | 0.90 |
| GAT | global | 0.20 | 0.04 | 0.31 | 0.28 | 0.25 | 0.47 | 0.71 | 0.51 | 0.48 |
| | global(0.9) | 0.75 | 0.64 | 0.43 | 0.70 | 0.63 | 0.44 | 0.47 | 0.74 | 0.89 |
| | oracle | 0.70 | 0.62 | 0.34 | 0.70 | 0.66 | 0.63 | 0.62 | 0.74 | 0.80 |
| | approx | 0.68 | 0.58 | 0.32 | 0.70 | 0.64 | 0.62 | 0.57 | 0.68 | 0.79 |
| SAGE | global | 0.27 | 0.45 | -0.28 | -0.32 | -0.36 | -0.09 | 0.50 | 0.34 | 0.18 |
| | global(0.9) | 0.78 | 0.75 | 0.55 | 0.90 | 0.61 | 0.40 | 0.53 | 0.66 | 0.79 |
| | oracle | 0.76 | 0.88 | 0.55 | 0.87 | 0.75 | 0.31 | 0.25 | 0.68 | 0.69 |
| | approx | 0.76 | 0.88 | 0.54 | 0.86 | 0.73 | 0.33 | 0.23 | 0.68 | 0.68 |

