# OpenReview forum: "Generalization Bound for GNNs on Transductive Node Classification: A View from Optimal Transport"
_ICLR.cc/2026/Conference — Submitted to ICLR 2026_

### Official Review · Reviewer_VDoG · 2025-10-24

**Soundness:** 3
**Presentation:** 2
**Contribution:** 3
**Rating:** 4
**Confidence:** 2

**Summary:**

This paper targets transductive node classification and derives two 1-Wasserstein–based generalization bounds: a global bound that upper-bounds test 0–1 error by the training $\gamma$-margin loss plus the $W_1$ distance between train/test embedding distributions, and a class-wise bound that decomposes the generalization gap into class-proportion mismatch and within-class concentration terms with a high-probability statement. For SGC, the paper also presents an upper bound with a depth-decaying additive term governed by the nontrivial spectral radius, and evaluates rank correlations between the proposed quantities and empirical generalization across 5 GNNs and 9 datasets, alongside an RC baseline.

**Strengths:**

- The theory is constructed directly in the transductive setting, with the bound formed by $W_1$ between train/test embedding distributions and margin terms, aligning assumptions with the task and aiding interpretability and applicability.
- For SGC, the paper provides an upper bound with a depth-decaying additive term governed by the nontrivial spectral radius, offering a quantitative expectation of depth effects without relying on hard-to-obtain global Lipschitz constants.
- The empirical study covers SGC, GCN, GCNII, GAT, and GraphSAGE on nine datasets, and reports rank correlations alongside an RC baseline, enabling side-by-side comparison and interpretation.

**Weaknesses:**

- The evaluation focuses on rank correlation; the paper does not provide numerical calibration/coverage plots or statistics, making it hard to assess bound tightness.
- In the main experiments for Theorem 4.1, the 0.9-quantile of $M(f,\varphi)$ is used instead of the theoretical maximum; this deviates from the theorem, and its impact is not quantified in the main text.
- Baselines are limited (primarily RC), constraining breadth; while correlations are positive in many settings, they are not positive in all (model × dataset) combinations, and analyzing negative cases would be informative.

**Questions:**

- Could the main text include side-by-side results or a sensitivity analysis for using the 0.9-quantile of $M(f,\varphi)$ versus the theoretical maximum, to assess the consequences of deviating from the theorem?
- Could you add scatter plots of “bound value” vs. “test error” and coverage statistics to evaluate numerical calibration and tightness?
- For the class-wise approx bound and $W_1$ estimation, could you summarize key steps and computational costs to clarify the accuracy–efficiency trade-off and aid reproducibility?

---

> ### Author Response · Authors · 2025-11-19
>
> We deeply appreciate the thoughtful comments and constructive feedback. In particular, it is very encouraging that the review highlights (i) the interpretability and applicability of our transductive setting and Wasserstein-based bounds, (ii) the bound for SGC is expressed via a depth-decaying term that is easy to compute, and (iii) the strength of the empirical results. Below, the comments and questions are addressed in detail.
>
> > W1. The evaluation focuses on rank correlation; the paper does not provide numerical calibration/coverage plots or statistics, making it hard to assess bound tightness. / Q2. Could you add scatter plots of “bound value” vs. “test error” and coverage statistics to evaluate numerical calibration and tightness?
>
> We provide scatter plots of test error vs. bound value for SGC model in Cora and CiteSeer datasets in the [anonymous link](https://anonymous.4open.science/api/repo/iclr2025-scatter-plot-20AB/file/test_bound_with_caption.pdf?v=140e66f9).
>
> Regarding coverage and tightness: in our experiments, the proposed bound covers almost all test errors, and we acknowledge that this behavior arises because the bound is numerically loose rather than tight. Our primary goal, however, is to provide a meaningful generalization signal across models and datasets, which is supported through the strong rank correlations, rather than the tightness of the bounds.
>
> This perspective is consistent with recent work on GNN generalization bounds, which typically focuses less on absolute tightness with coverage and more on whether the components of the bound (e.g., number of iterations, layers, nodes) exhibit the same trends as the test error [1, 2, 3, 4].
>
> > W2. In the main experiments for Theorem 4.1, the 0.9-quantile of $M(f,\phi)$ is used instead of the theoretical maximum; this deviates from the theorem, and its impact is not quantified in the main text. / Q1. Could the main text include side-by-side results or a sensitivity analysis for using the 0.9-quantile of $M(f,\phi)$ versus the theoretical maximum, to assess the consequences of deviating from the theorem?
>
> We use a percentile of $M(f,\phi)$ as heuristic, but practical choice for improving robustness, rather than a modification of the theoretical inequality itself. The results with maximum of $M(f,\phi)$ are reported in Appendix D, and they show that the correlation becomes less stable across settings. We observe that the maximum is often dominated by extreme outliers in $(i,j,y)$, which makes the bound overly loose.
> To obtain a more robust estimate, we use the 0.9-percentile. This percentile-based choice removes extreme outliers while preserving the overall scale, and consistently yields more stable correlations across datasets and GNN architectures.
>
> > W3-1. Baselines are limited (primarily RC), constraining breadth.
>
> To further address the concern regarding baselines, during the rebuttal period we will (i) more represent quantitative comparisons, if prior work is comparable and aligned with ours and (ii) for works where a direct comparison is not natural due to differing assumptions or problem setups, we will briefly explain why a comparasion is not appropriate.
>
> > W3-2. While correlations are positive in many settings, they are not positive in all (model × dataset) combinations, and analyzing negative cases would be informative.
>
> We agree that correlations are not positive for all model×dataset combinations, and we appreciate the suggestion to examine these cases. In our experiments, only the GCNII model on the Chameleon dataset exhibits a negative rank correlation. We analyse the result but do not find a consistent structural pattern that would support a strong, general claim about its cause, and we therefore refrain from making speculative interpretations based on this single case.
>
> [1] Oono, Kenta, and Taiji Suzuki. "Optimization and generalization analysis of transduction through gradient boosting and application to multi-scale graph neural networks." Advances in Neural Information Processing Systems 33 (2020): 18917-18930.
>
> [2] Esser, Pascal, Leena Chennuru Vankadara, and Debarghya Ghoshdastidar. "Learning theory can (sometimes) explain generalisation in graph neural networks." Advances in Neural Information Processing Systems 34 (2021): 27043-27056.
>
> [3] Cong, Weilin, Morteza Ramezani, and Mehrdad Mahdavi. "On provable benefits of depth in training graph convolutional networks." Advances in Neural Information Processing Systems 34 (2021): 9936-9949.
>
> [4] Tang, Huayi, and Yong Liu. "Towards understanding generalization of graph neural networks." International Conference on Machine Learning. PMLR, 2023.

---

> > ### Author Response · Authors · 2025-11-19
> >
> > > Q3. For the class-wise approx bound and $W_1$ estimation, could you summarize key steps and computational costs to clarify the accuracy–efficiency trade-off and aid reproducibility?
> >
> > The key steps for computing the expectation terms in Theorem 4.2 are as follows:
> >
> > 1. Sample a permutation $\pi'$ independently from the uniform distribution over all permutations.
> > 2. Using only the labels of the original training set (indexed by $\mathcal{I}_{\text{train}}^{(\pi)}$), construct class-wise index sets for $\pi'$.
> > 3. Approximate the quantities inside the expectation in Theorem 4.2 using constructed index set in 2.
> > 4. Repeat Steps 1–3 for $T$ times and approximate the expectations by empirical averages, $E_{\pi'} [\cdot] \approx \frac{1}{T}\sum_{t=1}^{T} (\cdot)$.
> >
> > (*We intended to provide detailed mathematical expressions for the index sets constructed in step 2, but we were unable to do so due to persistent equation rendering errors in Markdown. Instead, we have included the expressions in Appendix C of our paper, so please refer to that section.*)
> >
> > For a fixed dataset, the total runtime scales linearly with $T$. We use $T=4$ because larger sample sizes (16 or 64) do not provide consistent gains, while using smaller sample sizes ($1$) sometimes yields noticeably lower correlation. The corresponding correlation results and runtimes for SGC and GCN on Cora and Amazon-Ratings (with $1$, $4$, $16$, and $64$ samples) are reported in the tables below, and have been incorporated into the revised manuscript (please see Appendix C).
> >
> > * SGC-oracle
> >
> > | # of sampling | 1 | 4 | 16 | 64|
> > | -------- | -------- | -------- | -------- | -------- |
> > | Cora     |   0.84 (0.50s)  |   0.89 (1.99s) |   0.9 (8.22s)  |   0.88 (23.7s)  |
> > | Amazon-ratings     |   0.76 (9.69s)   |   0.93 (39.8s)  |   0.89 (147s)  |   0.91 (611s)  |
> >
> > * SGC-approx
> >
> > | # of sampling | 1 | 4 | 16 | 64|
> > | -------- | -------- | -------- | -------- | -------- |
> > | Cora     |   0.84 (0.36s)  |   0.87 (1.67s)   |   0.87 (6.21s)  |    0.86 (20.1s)  |
> > | Amazon-ratings     |   0.76 (0.83s)  |   0.91 (3.31s)  |   0.87 (13.2s)  |   0.92 (49.2s)   |
> >
> > * GCN-oracle
> >
> > | # of sampling | 1 | 4 | 16 | 64|
> > | -------- | -------- | -------- | -------- | -------- |
> > | Cora     |   0.7 (0.69s)  |   0.81 (2.58s)   |   0.75 (10.5s)   |   0.78 (43.9s)  |
> > | Amazon-ratings     |   0.9 (11.4s)  |   0.91 (45.0s)  |   0.88 (180s)  |   0.94 (728s)  |
> >
> > * GCN-approx
> >
> > | # of sampling | 1 | 4 | 16 | 64|
> > | -------- | -------- | -------- | -------- | -------- |
> > | Cora     |   0.67 (0.17s)   |   0.78 (0.68s)  |   0.72 (2.84s)  |    0.77 (12.1s)  |
> > | Amazon-ratings     |   0.88 (0.72s)  |   0.91 (2.89s)  |   0.87 (11.5s)   |   0.94 (46.8s)  |

---

> > > ### Author Response · Authors · 2025-11-29
> > >
> > > As noted in our earlier response, we have prepared a more detailed clarification regarding the baselines, and posted this as a general comment.
> > > We would appreciate it if the additional clarification could be reviewed.

---

### Official Review · Reviewer_6aQd · 2025-10-24

**Soundness:** 3
**Presentation:** 3
**Contribution:** 2
**Rating:** 4
**Confidence:** 3

**Summary:**

The paper extends the generalization error bounds established in arxiv:2106.03314 to the setting of transductive node classification. It derives two bounds expressed in terms of the Wasserstein distance. The first one (Theorem 4.1) demonstrates that the generalization error is small when the distance between the training and test encoded feature distributions is small. The second bound (Theorem 4.2) shows the generalization error is small when the distance between the training and test feature distributions within each class is small (which is analog to the results of Theorem 2 of arxiv:2106.03314) and when the difference in class proportions between the training and test sets across all classes is small. Finally, the paper analyzes the effect of depth on the generalization of GNNs. This section provides the only insight specific to GNNs, showing that the concentration and separation of node features play a crucial role in their generalization behavior.

**Strengths:**

1) The analysis is for node classification task which is not common in the literature. While there are existing bounds for node classification, many of them have limited practical utility and only hold for some architectures and training procedures. The bound here applies to a broader class of GNN architectures, and it is advantageous that it can be computed easily.

2) The provided bounds can be directly computed from the GNN embeddings, making them straightforward to apply in experiments.

**Weaknesses:**

1) Theorem 4.1 and 4.2 do not provide any GNN-specific insight; they lift the results from arxiv:2106.03314 to the transductive setting without yielding a deeper understanding of architectural behavior. Also see questions below.

2)  Theorem 6.1 is the only result that offers architectural insight, but it only applies to Simple Graph Convolution (i.e., a linear GNN). The restriction to linear GNNs should be stated clearly in the introduction, together with an explanation of why results obtained in this simplified setting still carry relevance for practice.


Comments regarding the structure
- typo in line 175: classidfier
- Dimension of the features: in line 138 X has dimension NxF, in line 183 Nxd
- Possible typo in Theorem 4.1: y should not have index i
- Vague reference in line 251: there are two terms with expectation in the bound, it is not clear which one referenced in the first sentence.
- Missing defination in line 267: Lip(ρf(·,c)) is not defined. It relies on the definition provided in arxiv:2106.03314, but this should be stated explicitly for clarity.

**Questions:**

1) What is the main challenge in extending the results of arxiv:2106.03314 to the transductive setting, and in what ways does the presented proof differ from those in arxiv:2106.03314?

2) The introduction claims that the Wasserstein-based bounds provide new theoretical insights (line 84). However, the conditions stated for reducing the generalization gap are all well established, including large margin of the classifier (arxiv:2106.03314), small distance of features within each class (arxiv:2106.03314) and the Lipschitz constant of f being small (arxiv:1706.08498). The insight regarding the oversmoothing is also not new since the same insight is discussed in arxiv:2205.12156, though examined using different tools.
What, concretely, is the new theoretical insight gained from the Wasserstein-based bound?

3) Contribution of the weights to the defination of the encoder: In line 376, the encoder is defined as

$\phi^{(l)}(x_i;X,A)=(A^l X)_{I}$

whereas in the description of the general setup (in lines 172-174), the encoder $\phi$ said to map to the embedding space which should be

$\phi^{(l)}(x_i;X,A)=(\sigma(AX^{(l)}W^{(l)}))_{I}$

 (lines 147-147) for a general GNN. The contribution of the weight could only be gathered in the classifier when considering a SGC. This raises the question of whether Theorems 4.1 and 4.2 are in fact derived only for a linear GNNs. If that is not the case, it is unclear where the contribution of the weights is captured in the analysis.

4) Which GNNs can be considered within this framework?

5) In the experiments the bound is only compared to the bounds in arxiv:2112.03968 (line 314) for GCN. The justification given is that other bounds are derived for specific architectures. However, GCN is already one of the simplest GNN architectures, and even the paper cited as arxiv:1905.10947 (in line 318) uses GCN. Could the authors clarify why the bound cannot be compared to these other results?

---

> ### Author Response · Authors · 2025-11-19
>
> We deeply appreciate the constructive feedback and recognition of our contribution in providing a practically useful, easily computable generalization bound that applies to most GNN architectures, which prior work had not achieved. Below, the remaining comments and concerns are addressed.
>
> > W1. Theorems 4.1 and 4.2 do not provide any GNN-specific insight; they lift the results from [1] to the transductive setting without yielding a deeper understanding of architectural behavior.
>
> We agree that Theorems 4.1–4.2 are not tied to any specific GNN architecture. However, they are applicable across all GNN architectures, and we view this as a strength rather than a limitation, because these theorems play a foundational role. A standard way to study the relationship between architecture and generalization is to first examine how different architectural choices affect the terms of an architecture-agnostic bound. In this sense, **having a practical, architecture-agnostic generalization bound is a prerequisite for understanding generalization behavior for various GNN architectures.**
>
> However, prior architecture-agnostic bounds require unrealistic subgraph-independence assumptions when applied to GNN node classification [7], or hard to compute and often poorly correlated with empirical generalization [1]. **Our bounds avoid these limitations and thus provide a more suitable and theoretically grounded basis for investigating how GNN architecture affects generalization.**
>
>
> > W2. Theorem 6.1 is the only result that offers architectural insight, but it only applies to Simple Graph Convolution (i.e., a linear GNN). The restriction to linear GNNs should be stated clearly in the introduction, together with an explanation of why results obtained in this simplified setting still carry relevance for practice.
>
> We agree that Theorem 6.1 offers architectural insight limited to SGC. However, we consider our work still meaningful since the combination of Theorem 4.2 and Theorem 6.1 provides, to our knowledge, the first generalization-based analysis that simultaneously reveals both the benefits and drawbacks of increasing the number of aggregations in GNN node classification. Since aggregation over graph neighbors is the key structural operation that distinguishes GNNs from other models, understanding how the number of aggregations affects performance is a central question underlying phenomena such as oversmoothing and oversquashing.
>
> In our work, Theorem 4.2 first establishes a generalization error bound that applies to general GNN architectures, and Theorem 6.1 then shows how this bound can be made explicitly dependent on the number of aggregation steps in the linear SGC case. Although this extension is restricted to linear GNNs, we view it as a principled starting point for extending such analysis to more expressive architectures like GCN or GAT for practice, and as an essential first step toward systematically linking depth and generalization across different GNN variants.
>
>
> > W3. Comments regarding the structure: typo in line 175: classidfier / Dimension of the features: in line 138 X has dimension NxF, in line 183 Nxd / Possible typo in Theorem 4.1: y should not have index i / Vague reference in line 251: there are two terms with expectation in the bound, it is not clear which one referenced in the first sentence. / Missing defination in line 267: Lip(ρf(·,c)) is not defined. It relies on the definition provided in [1], but this should be stated explicitly for clarity.
>
> Thank you for the detailed structural and editorial comments to improve our manuscript. We have corrected all typos, unified notations, and clarified ambiguity in the revised manuscript.
>
> Regarding Theorem 4.1, the label $y_i$ should have index, since it corresponds to the label of node $x_i$. In contrast, the unindexed y denotes a generic class label associated with a test node $x_j$, whose true label is unknown in the transductive setting; here $y$ is used to represent an arbitrary class, and thus does not carry an index.
>
>
> > Q1. What is the main challenge in extending the results of [1] to the transductive setting, and in what ways does the presented proof differ from those in [1]?
>
> The main challenge is that we cannot reuse several key assumptions and lemmas from [1], which are developed under an **i.i.d. sampling framework with a parametric assumption on the test data distribution**. In particular, the assumptions in [1] are not directly applicable to node classification, because in transductive node classification, we have access to edge information and test-node features before observing test labels, which breaks the independence structure required in [1].
>
> Technically, starting from **Theorem 4.2** (see Eq. (15) in Appendix A.2), our proof therefore diverges from [1]: we instead rely on **Definition A.2** and **Lemma A.3**, which are specifically tailored to the transductive case and the dependency structure induced by the graph.

---

> > ### Author Response · Authors · 2025-11-19
> >
> > > Q2. The introduction claims that the Wasserstein-based bounds provide new theoretical insights (line 84). However, the conditions stated for reducing the generalization gap are all well established, including large margin of the classifier ([1]), small distance of features within each class ([1]) and the Lipschitz constant of f being small ([2]). The insight regarding the oversmoothing is also not new since the same insight is discussed in [3], though examined using different tools. What, concretely, is the new theoretical insight gained from the Wasserstein-based bound?
> >
> > We can summarize our new theoretical insights as below:
> > * **(a)** The generalization error in the transductive setting is small when the feature distributions of the training and test sets are close (Theorem 4.1).
> > * **(b)** Strong intra-class concentration and inter-class separation in the representation space are essential for good generalization (Theorem 4.2).
> > * **(c\)** (Section 6) In the perspective of the generalization error, the optimal choice of depth should consider:
> >     * the trade-off induced by aggregation (since greater depth tightens the bound from the perspective of concentration but loosens it from the perspective of separation)
> >     * and the homophilic ratio of the graph.
> >
> > Regarding (b), prior work [1] indeed shows similar insights. However, **Theorem of [1] is proved only in an i.i.d. setting.** Since this assumption does not hold for node classification with GNNs, **our work is, to our knowledge, the first to rigorously justify the role of representation geometry in a transductive setting.** (As discussed in our answer to Q1, extending the results of [1] to GNNs is non-trivial.)
> >
> > Regarding (c\), our depth-related results and insights for node classification differ substantially from those in [3]:
> > * [3] analyzes depth from **a test MSE perspective** rather than from the generalization error.
> > * Their analysis **assumes training with ridge-regularized MSE**, which limits applicability in real-world scenarios.
> > * The benefit of depth in [3] is attributed to noise reduction, and the drawback is explained only through oversmoothing in the infinite-depth.
> > * They **do not analyze how homophily affects the optimal depth** in the classification task.
> >
> >
> > > Q3. Contribution of the weights to the defination of the encoder: In line 376, the encoder is defined as $\phi^{(\ell)}(x_i;\mathbf{X}, \mathbf{A})=(\mathbf{A}^{\ell}\mathbf{X})_i$ whereas in the description of the general setup (in lines 172-174), the encoder $\phi$ said to map to the embedding space which should be $\phi^{(\ell)}(x_i;\mathbf{X}, \mathbf{A})=(\sigma(\mathbf{A}\mathbf{X}^{(\ell)}\mathbf{W}^{(\ell)}))_i$ (lines 147-147) for a general GNN. The contribution of the weight could only be gathered in the classifier when considering a SGC. This raises the question of whether Theorems 4.1 and 4.2 are in fact derived only for a linear GNNs. If that is not the case, it is unclear where the contribution of the weights is captured in the analysis.
> >
> > Theorems 4.1 and 4.2 are not restricted to linear GNNs; they are formulated to apply to any GNN encoder. Please also see our response to W1. In our framework, the contribution of the weights is captured implicitly through the encoded feature space: all Wasserstein terms are computed on the embeddings $\phi(\mathbf{x}_i;\mathbf{X},\mathbf{A})$, where $\phi$ already includes the effect of message passing, nonlinearities, and learned weights.
> >
> > In contrast, the definition $\phi^{(\ell)}(\mathbf{x}_i;\mathbf{X}, \mathbf{A})=(\mathbf{A}^{\ell}\mathbf{X})_i$ is specific to the SGC case and is used only in the depth analysis of Theorem 6.1. There, we restrict to a linear GNN to show that the asymptotic behavior of the Wasserstein term with depth aligns with oversmoothing theories. This illustrative SGC result does not limit the generality of the bounds in Theorems 4.1 and 4.2, which continue to hold for general (nonlinear, weighted) GNN architectures.

---

> > > ### Author Response · Authors · 2025-11-28
> > >
> > > > Q4. Which GNNs can be considered within this framework?
> > >
> > > **Our framework accommodates general GNN architectures, which is one of the main strengths of Theorems 4.1 and 4.2.** In particular, Theorems 4.1 and 4.2 are formulated to apply to any GNN encoder used to produce node embeddings, so they are not restricted to a specific GNN variant. We can validate this by evaluating correlation across a diverse set of GNNs, including SGC, GCN, GCNII, GAT, GraphSAGE, in Table 1.
> > >
> > > > Q5. In the experiments the bound is only compared to the bounds in [4] (line 314) for GCN. The justification given is that other bounds are derived for specific architectures. However, GCN is already one of the simplest GNN architectures, and even the paper cited as arxiv:1905.10947 (in line 318) uses GCN. Could the authors clarify why the bound cannot be compared to these other results?
> > >
> > > To further address the concern regarding baselines, during the rebuttal period we will (i) more represent quantitative comparisons, if prior work is comparable and aligned with ours and (ii) for works where a direct comparison is not natural due to differing assumptions or problem setups, we will briefly explain why a comparasion is not appropriate.
> > >
> > > *In line 314, the reference was incorrectly assigned (we cited a different paper by the same authors from another year). This has been corrected.*
> > >
> > > [1] Chuang, Ching-Yao, et al. "Measuring generalization with optimal transport." Advances in neural information processing systems 34 (2021): 8294-8306.
> > >
> > > [2] Bartlett, Peter L., Dylan J. Foster, and Matus J. Telgarsky. "Spectrally-normalized margin bounds for neural networks." Advances in neural information processing systems 30 (2017).
> > >
> > > [3] Keriven, Nicolas. "Not too little, not too much: a theoretical analysis of graph (over) smoothing." Advances in Neural Information Processing Systems 35 (2022): 2268-2281.
> > >
> > > [4] Esser, Pascal, Leena Chennuru Vankadara, and Debarghya Ghoshdastidar. "Learning theory can (sometimes) explain generalisation in graph neural networks." Advances in Neural Information Processing Systems 34 (2021): 27043-27056.

---

> > > > ### Author Response · Authors · 2025-11-29
> > > >
> > > > As noted in our earlier response, we have prepared a more detailed clarification regarding the baselines, and posted this as a general comment.
> > > > We would appreciate it if the additional clarification could be reviewed.

---

### Official Review · Reviewer_C8YK · 2025-10-31

**Soundness:** 3
**Presentation:** 3
**Contribution:** 2
**Rating:** 2
**Confidence:** 4

**Summary:**

The paper studies the generalization of GNNs in the transductive setting from an optimal transport view. In my view, the paper uses an interesting perspective, but it cannot be considered generalization bound given that it does not have the equation form that generalization considers.

**Strengths:**

The paper studies a very relevant topic, and does it from an interesting perspective. The paper is well written and the ideas well presented.

**Weaknesses:**

- The work overlooks a lot of recent work on this topic such as:
A Manifold Perspective on the Statistical Generalization of
Graph Neural Networks
Wang et al.

Covered Forest: Fine-grained generalization analysis of graph neural networks
Vasileiu et at.

And more works than can be found in the previous two works.

- I do not think that this framework is a relevant one to study generalization. Theorem 4.1 needs a better justification of why this problem is akin to a Supremum over the space of functions. In particular, the bound does not have a clear dependency upon the class of functions being considered.

- This work is more focused on changes on the label distribution than a generalization analysis. I believe the authors should make a better case comparing their solution to the generalization problem presented by Vapnik.

**Questions:**

See weaknesses.

---

> ### Author Response · Authors · 2025-11-19
>
> We sincerely appreciate the time and effort devoted to this review, as well as the positive comments on the writing. Below, the comments are addressed, and several potential misunderstandings are clarified.
>
> > W1. The work overlooks a lot of recent work on this topic such as: [1] A Manifold Perspective on the Statistical Generalization of Graph Neural Networks (Wang et al., 2025), [2] Covered Forest: Fine-grained generalization analysis of graph neural networks (Vasileiu et at., 2024), And more works than can be found in the previous two works.
>
> [2] analyzes **graph-level classification under an i.i.d. distribution** over graphs and derives bounds in that setting. Our work, in contrast, studies transductive node classification (test node features/edges known, labels unknown) with instance-level bounds. Since **[2] does not address the transductive node setting, it lies outside the scope of the problem** our paper focuses on.
>
> To further address the reviewer's concern regarding recent work including [1], we will (i) represent more quantitative comparisons, if prior work is comparable and aligned with ours, and (ii) for works where a direct comparison is not natural due to differing assumptions or problem setups, we will briefly explain why a comparison is not appropriate, during the rebuttal period.
>
> > W2. I do not think that this framework is a relevant one to study generalization. Theorem 4.1 needs a better justification of why this problem is akin to a Supremum over the space of functions. In particular, the bound does not have a clear dependency upon the class of functions being considered. / W3. This work is more focused on changes on the label distribution than a generalization analysis. I believe the authors should make a better case comparing their solution to the generalization problem presented by Vapnik.
>
> From the perspective of classical generalization theory, this line of work may initially appear to deviate from the traditional focus on hypothesis-class complexity. Over the past decade, researchers have found that standard capacity-based bounds often fail to predict the behavior of deep networks, producing vacuous estimates or contradicting empirical evidence in large-scale studies. To address these issues, influential work such as Spectrally-normalized Margin Bounds for Neural Networks [3], cited over 1600 times, proposed a margin-based framework that, similar to the present line of research, **evaluates generalization not at the level of an abstract hypothesis space but at the level of a trained model’s properties.** On the other hand, these limitations further motivated the creation of the Predicting Generalization in Deep Learning (PGDL) competition [4], which called for complexity measures that could reliably rank trained neural networks across architectures, hyperparameters, and datasets.
>
> The strong empirical performance of several representation-based predictors [5] in PGDL revealed that the geometry of learned features plays a critical role, even though such measures initially lacked rigorous theoretical foundations [6]. In response, subsequent research introduced new representation-aware approaches that reflect a broader historical shift toward data-dependent, empirically aligned generalization theory in deep learning [7]. Our work aligns with this trajectory by likewise emphasizing the structural properties of learned representations as a key lens for understanding and predicting generalization.
>
>
> [1] Wang, Zhiyang, Juan Cervino, and Alejandro Ribeiro. "A manifold perspective on the statistical generalization of graph neural networks." arXiv preprint arXiv:2406.05225 (2024).
> [2] Vasileiou, Antonis, et al. "Covered Forest: Fine-grained generalization analysis of graph neural networks." arXiv preprint arXiv:2412.07106 (2024).
> [3] Bartlett, Peter L., Dylan J. Foster, and Matus J. Telgarsky. "Spectrally-normalized margin bounds for neural networks." Advances in neural information processing systems 30 (2017).
> [4] Jiang, Yiding, et al. "Neurips 2020 competition: Predicting generalization in deep learning." arXiv preprint arXiv:2012.07976 (2020).
> [5] Natekar, Parth, and Manik Sharma. "Representation based complexity measures for predicting generalization in deep learning." arXiv preprint arXiv:2012.02775 (2020).
> [6] Jiang, Yiding, et al. "Methods and analysis of the first competition in predicting generalization of deep learning." NeurIPS 2020 Competition and Demonstration Track. PMLR, 2021.
> [7] Chuang, Ching-Yao, et al. "Measuring generalization with optimal transport." Advances in neural information processing systems 34 (2021): 8294-8306.

---

> > ### Comment · Reviewer_C8YK · 2025-11-26
> >
> > I read and reviewed the paper again. My opinions remain unchanged over the setting considered in the paper.

---

> ### Author Response · Authors · 2025-11-28
>
> Thank you for taking the time to re-read our paper.
> To better understand your perspective and to accommodate a broader range of opinions in future revisions, **could you please elaborate more on which aspects of our setting or analysis make you consider that it is not a relevant approach to generalization**?
> We would also appreciate **clarification on why this view remains unchanged even in light of the recent generalization trends** we highlighted in our rebuttal.
>  A more detailed explanation of your viewpoint would be very helpful for us.

---

> ### Author Response · Authors · 2025-11-29
>
> As noted in our earlier response, we have prepared a more detailed clarification regarding the baselines, and posted this as a general comment.
> We would appreciate it if the additional clarification could be reviewed.

---

### Official Review · Reviewer_nuEj · 2025-11-11

**Soundness:** 3
**Presentation:** 2
**Contribution:** 3
**Rating:** 4
**Confidence:** 3

**Summary:**

This paper introduces two generalization error bounds for the transductive node classification setting based on the Wasserstein distance. The global one ties the generalization bound to the Wasserstein distance between node feature distributions of the training and test sets, while the local one focuses on the intra-class Wasserstein distance between these sets.
Besides, the study theoretically and empirically analyzes the impacts of graph topology and propagation depth.
In the experimental section, the proposed two bounds are verified to be correlated with the empirical error gap on different GNN architectures for homophilic and heterophilic datasets.
In summary, this paper is logically coherent with a sound experimental design.
I think it would be more impactful if additional analysis were included on how the proposed bounds can guide GNNs toward achieving better performance.

**Strengths:**

1). The paper is logically coherent, proposing two generalization error bounds that link the bound magnitude to node feature distributions. Specifically, the bounds are associated with the Wasserstein distance between training and test sets, which offers a novel perspective.

2). Experiments validate the correlation between the two proposed generalization error bounds and the empirical error gap on both homophilic and heterophilic datasets, demonstrating the effectiveness and performance.

3.) Experiments  analyze and visualize the impacts of GNN layers and graph topology on the Wasserstein distance, which supports the rationality of the proposed bounds.

**Weaknesses:**

1). The notation definitions are redundant and cumbersome. For instance, the definitions of training and test datasets in Line 128 and Line 184 overlap, and the description of GNN’s aggregation of neighboring nodes is repeated. It is recommended to streamline the notation and definitions.

2). The paper lacks clarity in certain aspects: the model’s loss function is not explicitly specified, and additional analysis should be provided on how the proposed generalization error bounds can guide model improvement.

3). Additional weaknesses are provided in the Questions.

**Questions:**

1). How is $\pi'$ selected in Equation 2? How many times is it sampled in experiments?

2). Regarding the calculations of $\mathcal{I}_{\text{test},c}^{(\pi')}$ and $m_c^{(\pi')}$ in Theorem 4.2, does the computation of $\mathcal{W}_1$ involve test label leakage for oracle method?

3). What does $c'$ denote in Line 263? It is used without prior definition.

4). Why is the magnitude difference between oracle and approx values small in some datasets but large in others? For example, in Roman-empire with SAGE, both values are 0.68; in Computers with GAT, the oracle, approx, and global values are all 0.70. However, significant discrepancies exist in other datasets, such as Squirrel with GCN (0.42 vs. 0.38). Could you explain this phenomenon across different datasets and GNN models?

5). What is the loss function used in the experiments? How can Theorem 4.1 and Theorem 4.2 guide the design of the loss function?

6). From the experimental results, the global bound outperforms the oracle bound. Is there a magnitude relationship between the bounds in Theorem 4.1 and Theorem 4.2? Can it be derived that the bound in Theorem 4.1 is smaller than that in Theorem 4.2?

---

> ### Author Response · Authors · 2025-11-19
>
> We deeply appreciate the Reviewer for the time and effort devoted and recognizing (i) the novelty of the proposed Wasserstein distance–based generalization error bounds and (ii) that the experiments validate their effectiveness and performance. Below, the comments and questions are addressed in detail.
>
> > W1. The notation definitions are redundant and cumbersome. For instance, the definitions of training and test datasets in Line 128 and Line 184 overlap, and the description of GNN’s aggregation of neighboring nodes is repeated. It is recommended to streamline the notation and definitions.
>
> Thank you for the helpful editorial comment. We have revised the manuscript to remove the overlapping definitions about training and test datasets. If the comment about a repeated description of the GNN aggregation refers to $\phi(\mathbf{x};\mathbf{X},\mathbf{A})$ in line 176, this passage is intentionally rephrased briefly to clarify the dependence of $\mathbf{X}$ and $\mathbf{A}$.
> If a different part of the manuscript is meant instead, we would be grateful if this could be pointed out, and we will correct it to further improve the manuscript.
>
> > W2-1. The paper lacks clarity in certain aspects: the model’s loss function is not explicitly specified. / Q5-1. What is the loss function used in the experiments?
>
> We appreciate the opportunity to clarify the loss functions. Theorems 4.1 and 4.2 are **agnostic to the loss** used to learn $f\circ\phi$, while they bound the gap between 0-1 loss $R_u(f\circ\phi;\pi)$ and the $\gamma$-margin loss $R_{m,\gamma}(f\circ\phi;\pi)$ as defined in line 186-190 of the manuscript. In our experiments, we train $f\circ\phi$ with cross‑entropy, the standard classification loss. This generalization analysis framework with margin loss is common and can be found in [1, 2, 3, 4], among others.
>
> > W2-2. Additional analysis should be provided on how the proposed generalization error bounds can guide model improvement. / Q5-2. How can Theorem 4.1 and Theorem 4.2 guide the design of the loss function?
>
> Theorem 4.1 suggests that an loss function should encourage the train and test representation distributions to be aligned, while Theorem 4.2 indicates that good generalization arises when representations exhibit strong intra-class concentration and inter-class separation.
>
> These theorems therefore provide an implicit guideline for loss design, but they can also naturally point to more explicit constructions as future work. For example, a straightforward idea inspired by our result is to incorporate the Wasserstein distances in the theorems directly into the loss function. However, because the Wasserstein distance on discrete representations is not differentiable, a differentiable relaxation such as a Sinkhorn loss would be required. Developing such OT-based losses is an interesting direction for future work.
>
> In addition, the guidance from Theorem 4.2 is consistent with what cross-entropy implicitly promotes. Therefore, our theory also provides a conceptual justification for the widespread use of cross-entropy loss in node classification tasks.
>
> > Q1. How is $\pi'$ selected in Equation 2? How many times is it sampled in experiments?
>
> Each $\pi'$ is independently sampled $4$ times from the uniform distribution over all permutations.
>
> For further information, we also report the correlation results for SGC and GCN on Cora and Amazon-Ratings when varying the sampling number $(1,4,16,64)$ in the tables below. These results have been incorporated into the revised paper (see Appendix C).
>
> * SGC-oracle
>
> | # of sampling | 1 | 4 | 16 | 64|
> | -------- | -------- | -------- | -------- | -------- |
> | Cora     |   0.84  |   0.89 |   0.9  |   0.88  |
> | Amazon-ratings     |   0.76   |   0.93  |   0.89  |   0.91  |
>
> * SGC-approx
>
> | # of sampling | 1 | 4 | 16 | 64|
> | -------- | -------- | -------- | -------- | -------- |
> | Cora     |   0.84  |   0.87   |   0.87  |    0.86  |
> | Amazon-ratings     |   0.76  |   0.91  |   0.87  |   0.92   |
>
> * GCN-oracle
>
> | # of sampling | 1 | 4 | 16 | 64|
> | -------- | -------- | -------- | -------- | -------- |
> | Cora     |   0.7  |   0.81   |   0.75   |   0.78  |
> | Amazon-ratings     |   0.9  |   0.91  |   0.88  |   0.94  |
>
> * GCN-approx
>
> | # of sampling | 1 | 4 | 16 | 64|
> | -------- | -------- | -------- | -------- | -------- |
> | Cora     |   0.67   |   0.78  |   0.72  |    0.77  |
> | Amazon-ratings     |   0.88  |   0.91  |   0.87   |   0.94  |

---

> > ### Author Response · Authors · 2025-11-19
> >
> > > Q2. Regarding the calculations of $\mathcal{I}^{(\pi')}_{\text{test,c}}$ and $m^{(\pi')}_c$ in Theorem 4.2, does the computation of $\mathcal{W}_1$ involve test label leakage for oracle method?
> >
> > Yes. The *oracle* use test labels to get $\mathcal{I}^{(\pi')}_{\text{test,c}}$ and $m^{(\pi')}_c$. This is why we further propose *approx*, which approximates the bound using only training data without test label leakage. Therfore, *approx* illustrates an approximation for real setting sceanarios, while *oracle* validates the theory. Please see line 323-326 in the paper.
> >
> > > Q3. What does $c'$ denote in Line 263? It is used without prior definition.
> >
> > $c'$ denotes a class label distinct from $c$. Accordingly, $\mathcal{W}(\cdot,\cdot)$ in line 263 represents the Wasserstein‑1 distance between the embedding distributions of two different classes $c$ and $c'$. We have revised it in revised manuscript for clarity.
> >
> > > Q4. Why is the magnitude difference between oracle and approx values small in some datasets but large in others? For example, in Roman-empire with SAGE, both values are 0.68; in Computers with GAT, the oracle, approx, and global values are all 0.70. However, significant discrepancies exist in other datasets, such as Squirrel with GCN (0.42 vs. 0.38). Could you explain this phenomenon across different datasets and GNN models?
> >
> > We thank the reviewer for pointing out the difference between the oracle and approx results. In line with this comment, we attempt to analyze these cases further but do not identify a consistent explanation that would justify a claim about the source of the discrepancy. However, we note that, in cases such as Squirrel with GCN, the difference between the oracle and approx rank correlations (0.42 vs. 0.38) is numerically small. Given that each correlation is computed over roughly 72 configurations, this gap is well within the level of variation one would typically expect from finite-sample effects.
> >
> >
> > > Q6. From the experimental results, the global bound outperforms the oracle bound. Is there a magnitude relationship between the bounds in Theorem 4.1 and Theorem 4.2? Can it be derived that the bound in Theorem 4.1 is smaller than that in Theorem 4.2?
> >
> > There is no general magnitude ordering between the global bound (Theorem 4.1) and the class-wise bound (Theorem 4.2). We conjecture that the global bound can sometimes appear tighter for reasons:
> > (i) the derivation of Theorem 4.2 introduces additional upper-bounding steps due to the decomposition into class-wise terms, which can make it numerically looser and (ii) the class-wise bound becomes increasingly loose as the train–test class proportions diverge as shown in $\mathbb E_{\pi'}\left[\sum_{c=1}^K\biggl|\frac{u_c^{(\pi')}}{u}-\frac{m_c^{(\pi')}}{m}\biggl|\\right]$ in Theorm 4.2.
> >
> > Rather than focusing on a performance comparison between the two bounds, we would like to emphasize their complementary roles. Theorem 4.1 provides a fast, label-agnostic way to estimate an error bound, while Theorem 4.2 not only yields an error bound but also highlights that concentration and separation of feature distributions are key factors for generalization. Both bounds thus play useful and distinct roles within our framework.
> >
> > [1] El-Yaniv, Ran, and Dmitry Pechyony. "Transductive rademacher complexity and its applications." Journal of Artificial Intelligence Research 35 (2009): 193-234.
> >
> > [2] Chuang, Ching-Yao, et al. "Measuring generalization with optimal transport." Advances in neural information processing systems 34 (2021): 8294-8306.
> >
> > [3] Cong, Weilin, Morteza Ramezani, and Mehrdad Mahdavi. "On provable benefits of depth in training graph convolutional networks." Advances in Neural Information Processing Systems 34 (2021): 9936-9949.
> >
> > [4] Bartlett, Peter L., Dylan J. Foster, and Matus J. Telgarsky. "Spectrally-normalized margin bounds for neural networks." Advances in neural information processing systems 30 (2017).

---

### Author Response · Authors · 2025-11-29

**Baseline selection**

In responding to the reviewers’ concerns regarding baselines, we first distinguish between two lines of generalization error bounds:
* **(a)** foundational, model-agnostic transductive generalization bounds, and
* **(b)** architecture-specific analyses that build on (a).

Model-agnostic generalization bounds, (a), such as those in [1, 2, 3], based on Rademacher complexity, stability, and PAC-Bayesian theory, are formulated independently of any GNN-specific components and serve as general tools for reasoning about generalization in transductive setting. Many subsequent works on generalization error in GNNs, (b), then develop GNN-specific analyses by expressing the complexity measures in (a) depending explicitly on GNN architectural factors (e.g., depth, graph Laplacian eigenvalues, etc). For example, [4, 5, 6] upper-bound the Rademacher complexity in [1] using normalized adjacency matrices, diffusion operators, and Lipschitz constants, and [7] derives a depth-dependent bound using stability theory [2].

**Theorem 4.1 and 4.2 belong to category (a), and thus, our work should be compared to the line of work (a).** However, Rademacher complexity in [1] is not computable in practice. For this reason, we use [5] to to enable comparison between Theorem 4.1/4.2 and [1].

We also considered adopting transductive stability bound [2] as a baseline, but computing the required $\epsilon$ term is computationally too expensive ($O((m+u)^2)$). An indirect comparison through [7] was not feasible either, as [7] assumes a binary classifier in its analysis, which is incompatible with the multi-class classification setting used in our experiments.

Finally, **we additionally report transductive PAC-Bayesian–based bounds [3] as comparison baselines,** and include the results in Table 1 in revised manuscript. The PAC-Bayesian–based bound performs consistently only for SGC, but frequently shows negative correlation for other architectures, suggesting that it is not a generally reliable bound across GNN models.


**Advantages of Our Bounds over Existing Baselines**

Theorems 4.1 and 4.2 offer several advantages over existing baselines.
* First, they provide, to the best of our knowledge, **the first generalization bounds in the transductive setting that are both computationally tractable and consistently positively correlated with test error across diverse datasets and GNN architectures**. This makes them substantially more usable in practice than prior bounds, which are either non-computable (e.g., Rademacher complexity) or exhibit weak or negative correlation.
* Second, because our bounds are model-agnostic and representation-based, they can naturally serve as the foundations upon which architecture-specific analyses (the “(b)-line” work) can be built to explain real-world generalization behavior.
* Finally, Section 6 demonstrates this potential concretely: by instantiating our bounds in the SGC architecture, we obtain a depth–generalization trade-off and its dependence on graph homophily, insights that prior theoretical frameworks were unable to capture.

Together, these points highlight why Theorems 4.1 and 4.2 provide a stronger and more practically relevant basis for generalization analysis than existing baselines.

**Note**: We recognize that the current title may unintentionally underemphasize the core theoretical contribution of Theorems 4.1 and 4.2 by suggesting that they serve merely as a means toward Section 6. To more clearly reflect the main contributions of the paper, we are considering revising the title such as **"Optimal Transport–Based Transductive Generalization and Applications to Graph Node Classification"**.

---

### Meta-Review · Area_Chair_F7mj · 2026-01-06

**Summary:**

The paper provide a new generalization bound for transductive node classification. The concerns of the reviewers are summarized as follows.
1. Mathematical Exposition: The manuscript requires improvement in mathematical writing. Issues include unclear notation, redundancy, and a general lack of clarity that hinders comprehension.
2. Literature Review: The paper overlooks several key recent works. A detailed comparison with this contemporary literature is necessary to properly position the authors' contributions.
3. Problem Motivation: The scientific value and motivation for the specific problem setting are not sufficiently justified or compelling.
4. Technical Novelty: The technical novelty is limited. The core theoretical framework appears to be a direct extension of results from [1] to a transductive setting, without yielding new, deeper insights into architectural behavior.
5. Generality of Findings: The primary architectural insight provided is constrained to linear GNNs, severely limiting the generality and impact of the conclusion.

**Reviewer Concerns:**

All concerns are responded  by the authors. However, these weaknesses still outstanding.

**Reviewer Scores:**

4, would not change
2, would not change
4, would not change

---

### Decision · Program_Chairs · 2026-01-26

Reject